# FedHPro: Federated Hyper-Prototype Learning via Gradient Matching

**Huan Wang** [1 2]   **Jun Shen** [1 *]   **Haoran Li** [3]   **Zhenyu Yang** [4]   **Jun Yan** [1]   **Ousman Manjang** [5]   **Yanlong Zhai** [5]
**Di Wu** [6]   **Guansong Pang** [2 *]

## Abstract

Federated Learning (FL) enables collaborative training of distributed clients while protecting privacy. To enhance generalization capability in FL, prototype-based FL is in the spotlight, since shared global prototypes offer semantic anchors for aligning client-specific local prototypes. However, existing methods update global prototypes at the prototype-level via averaging local prototypes or refining global anchors, which often leads to semantic drift across clients and subsequently yields a misaligned global signal. To alleviate this issue, we introduce **hyper-prototypes**, defined by a set of learnable global class-wise prototypes to preserve underlying semantic knowledge across clients. The hyper-prototypes are optimized via gradient matching to align with class-relevant characteristics distilled directly from clients' real samples, rather than prototype-level descriptors. We further propose **FedHPro**, a Federated Hyper-Prototype Learning framework, to leverage hyper-prototypes to promote inter-class separability via mutual-contrastive learning with client-specific margin, while encouraging intra-class uniformity through a consistency penalty. Comprehensive experiments under diverse heterogeneous scenarios confirm that 1) hyper-prototypes produce a more semantically consistent global signal, and 2) FedHPro achieves state-of-the-art performance on several benchmark datasets. Code is available at https://github.com/mala-lab/FedHPro.

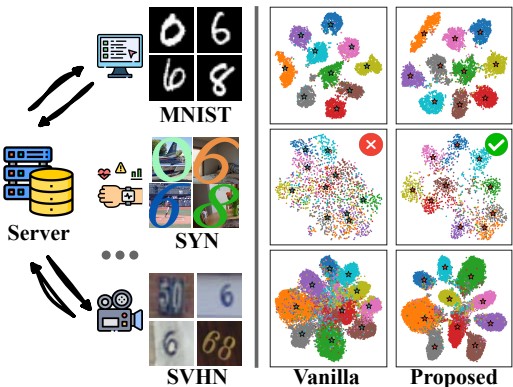

*Figure 1.* **Illustration of heterogeneous FL with domain skew.** The **Vanilla** column visualizes the feature distribution of a standard prototype-based FedProto (Tan et al., 2022a), showing failures in hard domains like SYN. In contrast, the **Proposed** column shows our approach achieves a larger inter-class distance and a smaller intra-class distance in such domains. Due to its simple aggregation, the prototypes in Vanilla (★) fail to reliably align with class centers, whereas our Proposed (★) yields better separability.

## 1. Introduction

Federated Learning (FL) has emerged as a paradigm to collaboratively train a global model across distributed clients based on heterogeneous data, without leaking their privacy (McMahan et al., 2017; Li et al., 2020a). However, one major challenge for FL is the data heterogeneity issue (Li et al., 2022), making the model convergence slow and unstable (Li et al., 2020d; Karimireddy et al., 2020). In the practical FL, each client's data is collected from highly diverse sources according to their own preference space. The non-independent and identically distributed (non-IID) data lead to inconsistency in local objectives among clients (Xu et al., 2025; Li et al., 2020b). Meanwhile, the global empirical direction is further distracted by aggregating these skewed local models, resulting in suboptimal performance.

To mitigate data heterogeneity, a mainstream of subsequent studies focuses on introducing various global signals to facilitate FL training (Li et al., 2020b; Mendieta et al., 2022; Luo et al., 2021). Considering practicality and decent generalization, prototype-based FL methods (Tan et al., 2022a; Yutong et al., 2023; Huang et al., 2023) have emerged as a novel FL paradigm that transfers global prototypes among clients to

---

[1] School of Computing and Information Technology, University of Wollongong, Wollongong, Australia [2] School of Computing and Information Systems, Singapore Management University, Singapore, Singapore [3] Monash University, Australia [4] Macquarie University, Australia [5] Beijing Institute of Technology, China [6] La Trobe University, Australia. Correspondence to: Jun Shen <jshen@uow.edu.au>, Guansong Pang <gspang@smu.edu.sg>.

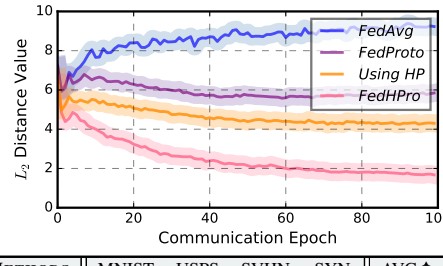

| METHODS | MNIST | USPS | SVHN | SYN | AVG ↑ | △ |
|---------|-------|------|------|-----|-------|---|
| FedAvg | 96.68 | 90.43 | 76.35 | 51.85 | 78.82 | – |
| FedProto | 97.40 | 91.89 | 79.22 | 53.15 | 80.41 | +1.59 |
| *Using HP* | 97.48 | 92.24 | 80.90 | 56.39 | **81.75** | **+2.93** |
| FedHPro | 98.52 | 93.13 | 84.95 | 62.59 | **84.80** | **+5.98** |

*Figure 2.* **Top**: The $L_2$ distance of centralized prototypes calculated by centralized training using all clients' samples to global prototypes from FedAvg (McMahan et al., 2017) and FedProto (Tan et al., 2022a), and from our hyper-prototypes. **Bottom**: The corresponding accuracy on the Digits (Peng et al., 2019) dataset, '*Using HP*' means replacing the global prototypes with our hyper-prototypes in the FedProto's loss function.

tackle data heterogeneity. They average the features with the same class to obtain local prototypes and upload them to the server to produce global prototypes. Afterward, the global prototypes are employed as signals to regularize local updating (Tan et al., 2022a). By aligning with global prototypes, they boost the generalizability (Zhou et al., 2025).

However, existing prototype-based FL methods directly collect the local prototypes from biased data distributions of clients, which unavoidably introduces semantic inconsistency (Zhou et al., 2023; 2025). This is evident in the feature distribution: FedProto (Tan et al., 2022a) shows clear separation between different categories in simple domains (*e.g.*, MNIST in Figure 1), but exhibits ambiguous clustering in hard domains (*e.g.*, SYN in Figure 1), particularly around samples near boundaries. Recent prototype-based FL methods, such as FedSA (Zhou et al., 2025), further improve global prototypes by learning or refining semantic anchors. However, their updates are still primarily performed at the prototype-level by client-specific representations, which may inherit semantic biases and thus fail to provide a consistently aligned global signal. We argue that a common weakness exists in these methods: *the class-wise semantic margins between global prototypes could be largely weakened by client-induced representation biases*. An intuitive solution is to preserve sufficient local data with diverse distributions to learn a uniform representation space, mirroring centralized training. However, this is fundamentally incompatible with the privacy-preserving FL paradigm.

This dilemma inspires our **core idea**: if the FL model could learn a representation space using all clients' samples, the obtained prototypes would exhibit better semantic consistency. In other words, we expect to optimize global prototypes over all real samples, instead of adopting the skewed

representations from clients. Inspired by this insight, we propose **hyper-prototypes**, a set of learnable global classwise prototypes optimized via gradient matching to explicitly align their gradients with the average gradients of real samples from clients. Such gradient alignment, by effectively simulating the optimization trajectories of real samples, enables our hyper-prototypes to distill class-relevant semantic characteristics directly from real samples, rather than prototype-level descriptors. This can be observed by a comparison of prototype generation methods in Figure 2: hyper-prototypes show a significantly smaller $L_2$ distance to centralized prototypes (calculated under centralized training with all clients' samples) compared to the prototypes from FedAvg (McMahan et al., 2017), and FedProto (Tan et al., 2022a). The advantages of our hyper-prototypes are further confirmed by the results, where our hyper-prototypes can be used to replace the prototypes in existing methods (*e.g.*, FedProto) to achieve largely enhanced performance.

Building on hyper-prototypes, we further propose **FedHPro**, a Federated Hyper-Prototype Learning framework for heterogeneous FL. FedHPro comprises two key components: 1) Hyper-Prototype Contrastive Learning (HPCL), which exploits the hyper-prototypes to construct mutual-contrastive learning (Chen et al., 2020; He et al., 2020; Gui et al., 2024) with a client-specific margin. HPCL first specifies the margin by measuring pair-wise distances between local prototypes to sharpen decision boundary. By pulling each embedding closer to its assigned hyper-prototypes and pushing it away from others, HPCL promotes inter-class separability while preserving semantics. 2) Hyper-Prototype Alignment Learning (HPAL), where we align each embedding with its representative hyper-prototypes by penalizing deviations at the feature level, thereby improving intra-class uniformity across clients. Our main contributions are listed as:

❶ We introduce hyper-prototypes (Section 2), a set of learnable global prototypes, optimized via gradient matching to align with class-relevant characteristics distilled from real samples, promoting semantic consistency across clients.

❷ We then propose FedHPro (Section 3), which leverages the hyper-prototypes to construct two complementary components: HPCL to enhance inter-class separability, while HPAL to impose intra-class uniformity.

❸ Extensive experiments across different FL scenarios demonstrate the effectiveness of FedHPro, and a series of ablative studies validate the advantages of hyper-prototypes.

## 2. Federated Learning with Hyper-Prototypes

### 2.1. Background and Motivation

**Preliminaries.** We focus on a standard heterogeneous FL setting (Li et al., 2020b; McMahan et al., 2017) with $K$ par-

ticipating clients holding data partition $\{\mathcal{D}_1, \mathcal{D}_2, \ldots, \mathcal{D}_K\}$. For the $k$-th client, its private data is $\mathcal{D}_k = \{x_i, y_i\}|_{i=1}^{n_k}$ and $y_i \in \{1, 2, \ldots, \mathbb{C}\}$, where $n_k$ denotes the scale of the client's dataset and $\mathbb{C}$ represents the number of classes. Let $n_k^c$ be the number of samples with label $c$ at the $k$-th client, so $\mathcal{D}_k^c = \{(x_i, y_i) \in \mathcal{D}_k | y_i = c\}$ represents the subset of data samples and $|\mathcal{D}_k^c| = n_k^c$, then the samples of label $c$ from all clients can be defined as $\mathcal{D}^c = \sum_{k=1}^{K} \mathcal{D}_k^c$ and $|\mathcal{D}^c| = n^c = \sum_{k=1}^{K} n_k^c$. Formally, the empirical objective of general FL can be formulated as follows:

$$w^* = \arg\min_w \mathcal{L}(w) = \sum_{k=1}^{K} \frac{n_k}{N} \mathcal{L}_k(w; \mathcal{D}_k), \quad (1)$$

where $\mathcal{L}_k$ is the local objective for the $k$-th client, $w$ and $N = \sum_{k=1}^{K} n_k$ denote the shareable model and the total number of samples from all clients, respectively.

Besides, participants agree on sharing a model with the same architecture. We regard the model with two modules: 1) a feature extractor $f : x \to z$, mapping each input sample $x$ to a $d$-dim feature vector $z = f(x)$; 2) a specific classifier $h : z \to \ell$, encoding the feature vector $z$ to a $\mathbb{C}$-dim logit output $\ell = h(z)$. The parameters of the $k$-th local model are denoted as $w_k = \{f_k, h_k\}$, and we further define the aggregated global model's parameters as $w^* = \{f^*, h^*\}$.

**Motivation.** The class-wise prototype $\mathbf{p}_k^c \in \mathbb{R}^d$ is calculated by the mean vector of the samples' features belonging to the same class $c$ on the $k$-th client, expressed as:

$$\mathbf{p}_k^c = \frac{1}{n_k^c} \sum_{(x_i, y_i) \in \mathcal{D}_k^c} f_k(x_i), \quad \mathbf{p}_k^c \in \mathbb{R}^d, \quad (2)$$

where $\mathcal{D}_k^c$ denotes the set of samples annotated with class $c$ on client $k$. The prototypes are typical of respective semantic knowledge, capturing useful class-relevant information on individual clients. We further calculate the $k$-th client's local prototypes, which are formulated as:

$$\mathcal{P}_k = [\mathbf{p}_k^1, \ldots, \mathbf{p}_k^c, \ldots, \mathbf{p}_k^{\mathbb{C}}] \in \mathbb{R}^{\mathbb{C} \times d}, \quad (3)$$

where $\mathbf{p}_k^c$ is the class-wise prototype defined in Equation (2), and $d$ indicates the vector dimension of the prototype. $\mathcal{P}_k$ is the group of prototypes for all categories on the $k$-th client.

Considering the large scale and inherent heterogeneity of participating clients in FL, simply using all clients' prototypes to form global signals is often insufficient. Several studies (Tan et al., 2022a;b) adopt a straightforward solution to obtain global prototypes by directly averaging all local prototypes. We follow (Tan et al., 2022a) to calculate the global prototypes $\mathbf{p}^c$ of class $c$ across clients:

$$\mathbf{p}^c = \frac{1}{|\mathcal{A}^c|} \sum_{k \in \mathcal{A}^c} \frac{n_k^c}{n^c} \mathbf{p}_k^c \in \mathbb{R}^d,$$
$$\mathbb{P} = [\mathbf{p}^1, \ldots, \mathbf{p}^c, \ldots, \mathbf{p}^{\mathbb{C}}] \in \mathbb{R}^{\mathbb{C} \times d}, \quad (4)$$

where $n^c$ is the number of samples belonging to class $c$ over all clients, and $\mathcal{A}^c$ means the set of clients that have class $c$. Then, the global prototypes $\mathbb{P}$ are dispatched as global signals to regularize local training (Tan et al., 2022b).

However, as mentioned above (Figure 1 and Figure 2), we observe that averaging local prototypes or refining global anchors may not capture desired information as expected. There are two notable problems: ❶ the single global prototype $\mathbf{p}^c$ is insufficient to describe the semantic information of class $c$ from different clients with biased preferences, and ❷ due to the inherent heterogeneity across clients, directly collecting local prototypes to form global prototypes would raise the same dilemma as the global drift, causing global prototypes to align with skewed representations.

### 2.2. Approximating Centralized Prototypes with Hyper-Prototypes via Gradient Matching

Driven by these limitations, we introduce hyper-prototypes, which 1) leverage a set of learnable class-wise prototypes to adequately capture semantically meaningful knowledge from each client on the server, and 2) optimize them through gradient matching to simulate optimization trajectories of real samples from clients (*i.e.*, approximating the prototypes that are calculated directly using all clients' samples).

For the $k$-th client, given the dataset $\mathcal{D}_k = \{x_i, y_i\}|_{i=1}^{n_k}$, we compute the average gradients $\mathbf{g}_k^c$ of the class $c$ as:

$$\mathbf{g}_k^c = \frac{1}{n_k^c} \sum_{(x_i, y_i) \in \mathcal{D}_k^c} \nabla_{z_i} \mathcal{L}_k(x_i, y_i) \in \mathbb{R}^d, \quad (5)$$

where $z_i = f_k(x_i)$ and the instance-wise loss is written as $\mathcal{L}_k(x_i, y_i)$, which can be instantiated by cross-entropy loss $\nabla_{z_i} \mathcal{L}_k(x_i, y_i) = (\frac{\partial h_k}{\partial z_i})^\top \nabla_{h_k} \mathcal{L}_{CE}(h_k(z_i), y_i)$, and $\mathbf{g}_k^c$ represents the gradients of the prototype $\mathbf{p}_k^c$ on the client $k$. Then, we can further obtain the gradients of all prototypes on the client $k$ as $\mathbf{g}_k = \{\mathbf{g}_k^1, \ldots, \mathbf{g}_k^c, \ldots, \mathbf{g}_k^{\mathbb{C}}\}$.

After the server receives the gradients from participating clients, we first aggregate these gradients for each class $c$ by averaging over all participating clients:

$$\mathbf{g}^c = \frac{1}{|\mathcal{A}^c|} \sum_{k \in \mathcal{A}^c} \mathbf{g}_k^c \in \mathbb{R}^d,$$
$$\mathcal{G} = [\mathbf{g}^1, \ldots, \mathbf{g}^c, \ldots, \mathbf{g}^{\mathbb{C}}] \in \mathbb{R}^{\mathbb{C} \times d}, \quad (6)$$

where $\mathcal{A}^c$ denotes the set of clients that hold class $c$. We initialize a set of learnable vectors $\{\mathbf{s}_i^c\}|_{i=1}^{|\mathcal{I}|} \in \mathbb{R}^{|\mathcal{I}| \times d}$ to simulate the hyper-prototypes of class $c$. Then, the global model calculates gradients of hyper-prototypes via a virtual loss function $\mathcal{L}_{vir}$, which can be formulated as:

$$\mathbf{g}_{\text{HP}}^c = \frac{1}{|\mathcal{I}|} \sum_{i=1}^{|\mathcal{I}|} \nabla_{\mathbf{s}_i^c} \mathcal{L}_{vir}(h^*(\mathbf{s}_i^c), y_{vir}) \in \mathbb{R}^d,$$
$$\mathcal{G}_{\text{HP}} = [\mathbf{g}_{\text{HP}}^1, \ldots, \mathbf{g}_{\text{HP}}^c, \ldots, \mathbf{g}_{\text{HP}}^{\mathbb{C}}] \in \mathbb{R}^{\mathbb{C} \times d}, \quad (7)$$

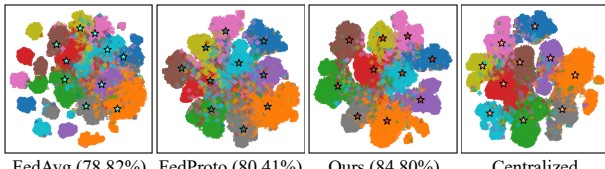

FedAvg (78.82%)  FedProto (80.41%)  Ours (84.80%)  Centralized

*Figure 3.* **Visualization for the representation space of different prototypes** on Digits (Peng et al., 2019). Each color (■ ■) indicates one class, and each shape (■ ●) denotes one domain. Centralized is trained on all clients' samples, as an upper-bound reference for prototype quality. The global prototypes of FedAvg and FedProto fail to describe diverse domain information, while our hyper-prototypes promote better semantic consistency across multiple domains. Please zoom in to view details.

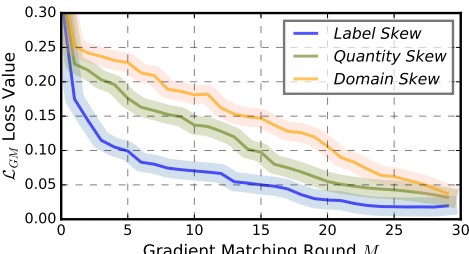

*Figure 4.* **Loss trend of** $\mathcal{L}_{GM}$ (Equation (8)) under different FL scenarios: CIFAR10 (Krizhevsky et al., 2009) with NID$1_{0.5}$ (label skew), CIFAR10-LT (Krizhevsky et al., 2009) with $\rho = 50$ (quantity skew), Digits (Peng et al., 2019) (domain skew).

where we set $\mathcal{L}_{vir}$ as the cross-entropy loss and $y_{vir}$ as the corresponding class label $c$, and $h^*$ denotes the classifier of the global model $w^*$. Based on $\mathbf{g}^c \in \mathcal{G}$ from real samples, we maximize the agreement between $\mathbf{g}^c$ and $\mathbf{g}^c_{\text{HP}}$ by measuring their dissimilarity to optimize hyper-prototypes:

$$\mathcal{L}_{GM}(\mathbf{g}^c, \mathbf{g}^c_{\text{HP}}) = 1 - \frac{\mathbf{g}^c \cdot \mathbf{g}^c_{\text{HP}}}{\|\mathbf{g}^c\|_2 \times \|\mathbf{g}^c_{\text{HP}}\|_2}, \quad (8)$$

where $\|\cdot\|_2$ means $L_2$ normalization. Through $\mathcal{L}_{GM}$ loss in Equation (8), after optimizing the hyper-prototypes on $M$ rounds, we can obtain the updated hyper-prototypes $\mathcal{S}_M$:

$$\mathcal{S}_M = [\{\mathbf{s}^1_i\}|_{i=1}^{|\mathcal{I}|}, \ldots, \{\mathbf{s}^c_i\}|_{i=1}^{|\mathcal{I}|}, \ldots, \{\mathbf{s}^{\mathbb{C}}_i\}|_{i=1}^{|\mathcal{I}|}], \quad (9)$$

where $\mathcal{S}_M$ denotes optimizing our hyper-prototypes over $M$ rounds, with each class $c$'s hyper-prototypes set as $\mathcal{S}^c_M = \{\mathbf{s}^c_i\}|_{i=1}^{|\mathcal{I}|} \in \mathbb{R}^{|\mathcal{I}| \times d}$ and $\mathcal{S}_M \in \mathbb{R}^{\mathbb{C} \times |\mathcal{I}| \times d}$. We also define $\mathcal{H}^c_M = \frac{1}{|\mathcal{I}|} \sum_{i=1}^{|\mathcal{I}|} \mathbf{s}^c_i \in \mathbb{R}^d$ as the averaged hyper-prototypes and $M$ as the iteration rounds. Note that the optimized $\mathcal{S}_M$ serves as the initialization for the next round of training.

### 2.3. Hyper-Prototypes vs. Prototypes

We further display the differences between the global prototypes and hyper-prototypes. As visualized in Figure 3 by t-SNE (Maaten & Hinton, 2008), global prototypes obtained from FedAvg (McMahan et al., 2017) or FedProto

(Tan et al., 2022a) inherently present limited class-wise information and show a representation space skewed toward the potentially dominant domains. Our hyper-prototypes instead achieve larger inter-class separations and tighter intra-class compactness, even across multiple domains.

Then, as shown in Figure 4, we further analyze the effectiveness of $\mathcal{L}_{GM}$ in Equation (8), which depicts its loss dynamics under various heterogeneous scenarios. It can be observed that the differences in terms of gradients decrease gradually after a few rounds, which means that optimizing $\mathcal{L}_{GM}$ successfully makes the gradients $\mathbf{g}^c_{\text{HP}}$ of our hyper-prototypes close to the $\mathbf{g}^c$ of the real samples, indicating a good approximation to the centralized prototypes.

Moreover, our hyper-prototypes can serve as a plug-and-play component to replace the conventional prototypes in existing methods to enhance their performance. As previously shown in Figure 2, the hyper-prototypes can be easily incorporated with the existing prototype-based FL methods to yield improvements by replacing the global prototypes, and more empirical results are provided in Table 3.

## 3. The Proposed FedHPro

After obtaining the hyper-prototypes $\mathcal{S}_M \in \mathbb{R}^{\mathbb{C} \times |\mathcal{I}| \times d}$, we further propose FedHPro, as shown in Figure 5, consists of two complementary modules: Hyper-Prototype Contrastive Learning (HPCL in Section 3.1) and Hyper-Prototype Alignment Learning (HPAL in Section 3.2). We also provide a convergence analysis of FedHPro in Section 3.3.

### 3.1. Hyper-Prototype Contrastive Learning

To maximally benefit local learning, inspired by the success of contrastive learning (He et al., 2020; Gui et al., 2024), we deem that a well-generalizable model should not only maintain a clear decision boundary across different categories, but also exhibit invariance among diverse samples with the identical semantics. Based on the hyper-prototypes $\mathcal{S}_M$ from the server, we devise Hyper-Prototype Contrastive Learning (HPCL) with client-specific margin to facilitate inter-class separability while preserving semantic meaning. Specifically, for $(x_i, y_i) \in \mathcal{D}^c_k$, we feed it into the feature extractor $f_k$ to get the embedding $z_i = f_k(x_i)$. Then, we calculate a margin among local prototypes for client $k$:

$$d_k = \frac{1}{(\mathbb{C}-1)^2} \sum_{c_1} \sum_{c_2 \neq c_1} D_{L_2}(\mathbf{p}^{c_1}_k, \mathbf{p}^{c_2}_k), \quad (10)$$

where $c_1, c_2 \in \{1, ..., \mathbb{C}\}$, $D_{L_2}(a, b)$ means the Euclidean distance between $a$ and $b$. The client-specific margin $d_k$ adaptively guides each sample's embedding toward a more compact cluster center. On this basis, we aim to enforce $z_i$ to be similar to respective $\mathcal{S}^c_M$ and dissimilar to the hyper-prototypes of other classes $\mathcal{N}^c_M = \{\mathcal{S}_M \setminus \mathcal{S}^c_M\}$. We define

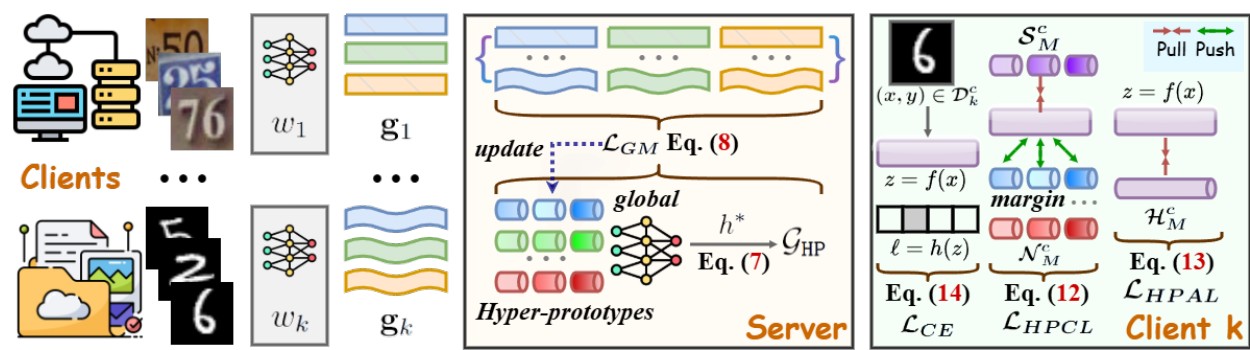

*Figure 5.* **Framework illustration** of Federated Hyper-Prototype Learning (FedHPro). The clients upload the gradients $\{\mathbf{g}_1, ..., \mathbf{g}_k\}$ (Equation (5), ▭◠) to the server. Based on these gradients from local clients, we leverage a set of learnable units (▭▭) to simulate hyper-prototypes, capturing class-relevant semantic properties from real samples via gradient matching ($\mathcal{L}_{GM}$ in Equation (8)) to enhance generalizability. Then, by incorporating the optimized hyper-prototypes ($\mathcal{S}_M$ in Equation (9)), we construct Hyper-Prototype Contrastive Learning (HPCL in Section 3.1) with client-specific margin to promote inter-class separability. Besides, we design Hyper-Prototype Alignment Learning (HPAL in Section 3.2) to impose intra-class uniformity by smoothly penalizing deviations at the feature-level.

the similarity between $z_i \in \mathbb{R}^d$ and $\mathcal{S}_M^c \in \mathbb{R}^{|\mathcal{I}| \times d}$ as:

$$s(z_i, \mathcal{S}_M^c) = \frac{1}{|\mathcal{I}|} \sum_{j=1}^{|\mathcal{I}|} \frac{z_i \cdot \mathbf{s}_j^c}{\|z_i\|_2 \times \|\mathbf{s}_j^c\|_2}. \quad (11)$$

Next, we introduce the following objective term to quantify and regulate local model training, formulated as:

$$\mathcal{L}_{HPCL} = \log(1 + \frac{\sum_{\mathcal{S}_M^j \in \mathcal{N}_M^c} \exp((s(z_i, \mathcal{S}_M^j) + d_k)/\tau)}{\exp(s(z_i, \mathcal{S}_M^c)/\tau)}), \quad (12)$$

where $\tau$ denotes a temperature parameter to control the representation strength (Chen et al., 2020). By minimizing the $\mathcal{L}_{HPCL}$ in Equation (12), the client's local model $w_k$ brings each sample's embedding $z_i$ closer to associated positive pair $\mathcal{S}_M^c$ ($\Rightarrow$ compactness) and away from other negative pairs $\mathcal{N}_M^c$ ($\Rightarrow$ exclusiveness), promoting inter-class separability in local model training.

### 3.2. Hyper-Prototype Alignment Learning

Despite the local model pulling sample's embedding toward the corresponding hyper-prototypes by minimizing $\mathcal{L}_{HPCL}$, representation inconsistency may still arise due to the intrinsic heterogeneity across clients, which fails to offer a stable convergence target. Therefore, we design Hyper-Prototype Alignment Learning (HPAL) to impose intra-class uniformity among clients by smoothly penalizing deviations at the feature-level. Given $(x_i, y_i) \in \mathcal{D}_k^c$ on the client $k$, we introduce a smooth regularization penalty term to pull the sample's embedding $z_i = f_k(x_i)$ closer to the averaged hyper-prototypes $\mathcal{H}_M^c = \frac{1}{|\mathcal{I}|} \sum_{i=1}^{|\mathcal{I}|} \mathbf{s}_i^c \in \mathbb{R}^d$. We minimize the discrepancy in a dimension-wise manner, with the loss

defined as follows (where $q$ indexes the dimension):

$$\mathcal{L}_{HPAL} = \sum_{q=1}^{d} \begin{cases} \frac{1}{2}(z_{i(q)} - \mathcal{H}_{M(q)}^c)^2; & \text{if } |z_{i(q)} - \mathcal{H}_{M(q)}^c| \leq 1, \\ |z_{i(q)} - \mathcal{H}_{M(q)}^c| - \frac{1}{2}, & \text{otherwise.} \end{cases} \quad (13)$$

We aim to smoothly align each embedding with its corresponding averaged hyper-prototypes, thereby imposing intra-class uniformity across clients. Besides, we utilize the sample's logit output $\ell_i = h_k(z_i)$ with original label $y_i$ to build the standard cross-entropy loss term as:

$$\mathcal{L}_{CE} = -\mathbf{1}_{y_i} \log(\delta(h_k(z_i))), \quad (14)$$

where $\delta$ means softmax function and $\ell_i = h_k(z_i) \in \mathbb{R}^{\mathbb{C}}$. Finally, we carry out the following objective for training:

$$\mathcal{L} = \mathcal{L}_{CE} + \mathcal{L}_{HPCL} + \mathcal{L}_{HPAL}. \quad (15)$$

In each communication round, local model $w_k$ is supervised by $\mathcal{L}$ (Equation (15)), then the server collects the gradients from clients to optimize hyper-prototypes and aggregates local models to produce the global model. The algorithm of the proposed FedHPro is provided in Algorithm 1.

### 3.3. Convergence Analysis of FedHPro

We prove the convergence of FedHPro in the non-convex case. For the non-convex and $L_1$-Lipschitz smooth objective function $\mathcal{L}$, there exists a constant $B > 0$ such that $\mathbb{E}_\xi[\|\nabla \mathcal{L}\|_2] \leq B$ for any sample $\xi$. We denote $\mathcal{L}$ in Equation (15) as $\mathcal{L}_r$ at round $r$, and all used assumptions are similar to general works (Li et al., 2020d;b; Ye et al., 2023).

**Theorem 3.1** (Non-convex convergence rate of FedHPro). *Let the communication round $r$ from $0$ to $R-1$, given any $\varepsilon > 0$, FedHPro will converge when,*

$$R > \frac{2}{E\eta} \cdot \frac{\mathcal{L}_0 - \min(\mathcal{L}^*)}{2\varepsilon - L_1\eta(\varepsilon + \sigma^2) - 2L_2B(\mathbb{C}-1)}, \quad (16)$$

*where* $\min(\mathcal{L}^*)$ *denotes the optimal solution of* $\mathcal{L}$.

Theorem 3.1 establishes that the expected $L_2$-norm of the gradients can be upper-bounded by an arbitrary $\varepsilon > 0$. The smaller $\varepsilon$ is, the larger $R$ is, which means that the tighter the bound is, the more communication rounds $R$ is required. Complete proofs are presented in Appendix Section B.

## 4. Related Work

**Heterogeneous Federated Learning.** Federated learning is proposed to collaboratively train a global model in a distributed learning environment, *e.g.*, Federated Averaging (FedAvg) (McMahan et al., 2017). However, its performance and convergence are impeded due to non-IID data, which further causes data heterogeneity in FL (Li et al., 2020a; Kairouz et al., 2021; Zhu et al., 2021). To mitigate the data heterogeneity, existing solutions mainly focus on two directions: regulating local training (Karimireddy et al., 2020; Li et al., 2020b; Wang et al., 2025) and optimizing the global model (Ye et al., 2023; Huang et al., 2022; Zhang et al., 2023). The former executes the adjustment on local training to decrease parameter differences across clients (Li et al., 2020b; Karimireddy et al., 2020). The latter aims to improve the performance of the global model at the server side (Qi et al., 2024; Ye et al., 2023; Chen et al., 2024). In addition, data augmentation (Yan et al., 2025; Ma et al., 2025; Wen et al., 2022) and knowledge distillation (Zhang et al., 2024b; Wang et al., 2023b) have emerged to bridge the gap between local distributions and the global one. Nevertheless, these methods only partially mitigate data heterogeneity and rely on auxiliary public data that may bring privacy risks.

**Prototype Learning.** A prototype is defined as the mean value of features with the same class semantics (Yang et al., 2018; Zhou et al., 2022; Tan et al., 2022a; Yutong et al., 2023). Due to its natural semantic characteristics and interpretability (Wang et al., 2023a; Nauta et al., 2023), it has shown strong potential (Li et al., 2024; Huang et al., 2025).

Recently, there has been growing interest in leveraging prototypes to mitigate data heterogeneity in FL (Tan et al., 2022a; Zhang et al., 2024b; Wang et al., 2025; Yutong et al., 2023; Zhou et al., 2025). The cornerstone prototype-based FL method, FedProto (Tan et al., 2022a), aggregates local prototypes collected from different clients to obtain global prototypes, and then sends the global prototypes back to all clients. FedTGP (Zhang et al., 2024a) uses an adaptive-margin-enhanced contrastive scheme to learn global prototypes, while FedSA (Zhou et al., 2025) procreates semantic knowledge across clients by their class prototypes.

However, these recent prototype-based FL methods (*e.g.*, FedSA (Zhou et al., 2025)) still update global prototypes mainly through prototype-level aggregation or anchor re-finement, which can inherit representation biases induced by heterogeneous clients. In contrast, we construct a set of hyper-prototypes as semantic anchors, linking class-relevant characteristics in each client. We optimize them through gradient matching on class-wise gradients distilled from real samples, without requiring access to any raw data.

**Contrastive Learning.** Contrastive learning has become a promising direction for self-supervised learning (He et al., 2020; Chen et al., 2020; Gui et al., 2024; Tian et al., 2020; Wang & Qi, 2022), achieving superior performance in vision transformers (Wang et al., 2022; Kim et al., 2023), graph mining (You et al., 2020; Xu et al., 2021), etc. Recent works are focused on how to select the positive and negative pairs via InfoNCE (Oord et al., 2019) and variants (Weinberger et al., 2023; Zeng et al., 2024). Another popular direction is to incorporate contrastive learning into the FL paradigm (Li et al., 2021; Tan et al., 2022b; Seo et al., 2024; Psaltis et al., 2023). In this work, contrastive learning is built upon the hyper-prototypes (via HPCL) with a client-specific margin to promote local training in heterogeneous FL.

## 5. Experiments

### 5.1. Experimental Setups

**Datasets and Models.** We consider popular datasets to cover medical, natural, and artificial scenarios: HAM10000 (Tschandl et al., 2018), CIFAR10 & CIFAR100 (Krizhevsky et al., 2009), TinyImageNet (Le & Yang, 2015), CIFAR10-LT, CIFAR100-LT, TinyImageNet-LT, Digits (Peng et al., 2019), and Office-Caltech (Gong et al., 2012). We use MobileNetV2 (Sandler et al., 2018) for TinyImageNet (LT), and ResNet-10 (He et al., 2016) for the others. All methods use the same architecture for fair comparison.

**Federated Scenarios.** 1) **Label Skew**: NID1$_\alpha$ follows Dirichlet distribution (Wang et al., 2020) ($\alpha$ as the non-IID level), NID2 is a more extreme setting consistting of 6 clients (each has a single class) and 1 client has all classes. 2) **Quantity Skew**: we shape the original data into a long-tailed distribution by (Kaidi et al., 2019), and $\rho$ means the ratio between sample sizes of the most frequent and lowest frequent class. 3) **Domain Skew**: Digits (Peng et al., 2019) includes 4 domains: MNIST, USPS, SVHN, SYN with 10 classes, and Office-Caltech (Gong et al., 2012) also consists of 4 domains: Caltech, Webcam, Amazon, DSLR with 10 classes. We set 20 and 10 clients and randomly assign these domains for Digits (M: 3, U: 7, SV: 6, SY: 4) and Office-Caltech (C: 3, W: 1, A: 2, D: 4), and each client's dataset is randomly sampled 1% (Digits) and 20% (Office-Caltech).

**Baselines.** We compare FedHPro with SOTA baselines, including standard FedAvg (McMahan et al., 2017), Fed-

*Table 1.* **Comparison results** under label and quantity skew. † denotes the results obtained by exchanging both the prototypes and model parameters (please see analysis in Appendix Table A7) during the FL training. Best results are in red bold, with second best in blue bold.

| METHODS | CIFAR10 | | | HAM10000 | | | TINYIMAGENET | | | CIFAR10-LT (NID1$_{0.5}$) | | |
|---|---|---|---|---|---|---|---|---|---|---|---|---|
| | NID1$_{0.2}$ | NID1$_{0.5}$ | NID2 | NID1$_{0.2}$ | NID1$_{0.5}$ | NID2 | NID1$_{0.2}$ | NID1$_{0.5}$ | NID2 | $\rho = 10$ | $\rho = 50$ | $\rho = 100$ |
| FedAvg [AISTATS'17] | 80.59 | 85.60 | 74.48 | 45.26 | 48.68 | 41.54 | 41.34 | 43.88 | 36.20 | 75.54 | 69.05 | 60.79 |
| FedProx [MLSys'20] | 80.74 | 85.54 | 74.67 | 45.42 | 48.55 | 41.21 | 41.16 | 43.50 | 36.62 | 75.15 | 69.23 | 60.86 |
| MOON [CVPR'21] | 82.04 | 86.95 | 76.07 | 47.33 | 50.24 | 43.10 | 43.22 | 44.36 | 37.15 | 76.11 | 69.70 | 60.91 |
| FedProto† [AAAI'22] | 81.61 | 86.25 | 75.32 | 46.35 | 49.70 | 42.45 | 42.25 | 44.19 | 36.86 | 75.81 | 69.43 | 60.37 |
| FedTGP† [AAAI'24] | 84.14 | 87.31 | **77.95** | 48.29 | 50.77 | 44.26 | 43.92 | 45.13 | 38.21 | 76.27 | 71.36 | 61.94 |
| FedGMKD [NeurIPS'24] | 83.56 | **88.09** | 77.29 | 47.97 | 50.49 | 44.20 | 44.09 | 45.30 | 37.94 | **76.83** | 71.21 | 62.13 |
| FedRCL [CVPR'24] | 83.80 | 87.76 | 77.51 | **48.60** | 51.10 | 44.36 | **44.30** | 45.24 | **38.76** | 76.60 | 71.55 | 62.41 |
| FedSA† [AAAI'25] | **84.27** | 87.93 | 77.86 | 48.45 | **51.28** | **44.85** | 44.18 | **45.56** | 38.35 | 76.75 | **72.30** | **62.48** |
| FedHPro (Ours) | **85.98** | **89.56** | **79.70** | **50.23** | **52.79** | **46.17** | **45.64** | **46.90** | **40.52** | **78.62** | **74.69** | **64.75** |

*Table 2.* **Comparison results** (domain skew), continuing Table 1.

| METHODS | OFFICE-CALTECH | | | | | |
|---|---|---|---|---|---|---|
| | Caltech | Webcam | Amazon | DSLR | AVG↑ | △ |
| FedAvg | 60.71 | 48.90 | 75.79 | 36.28 | 55.42 | – |
| FedProx | 60.77 | 50.83 | 77.27 | 37.61 | 56.62 | +1.20 |
| MOON | 57.75 | 46.08 | 72.83 | 34.50 | 52.79 | -2.63 |
| FedProto† | 61.58 | 53.21 | 78.72 | 39.53 | 58.26 | +2.84 |
| FedTGP† | 61.25 | 54.97 | 78.79 | 42.57 | 59.39 | +3.97 |
| FedGMKD | 61.78 | 55.36 | 79.30 | 43.22 | 59.91 | +4.49 |
| FedRCL | 60.29 | 50.41 | 74.54 | 37.46 | 55.68 | +0.26 |
| FedSA† | 62.23 | 55.85 | 79.13 | 45.05 | **60.57** | **+5.15** |
| FedHPro | 64.61 | 62.45 | 80.69 | 50.33 | **64.52** | **+9.10** |

Prox (Li et al., 2020b); contrastive-based MOON (Li et al., 2021), FedRCL (Seo et al., 2024); prototype-based FedSA (Zhou et al., 2025), FedProto (Tan et al., 2022a), FedTGP (Zhang et al., 2024a), and FedGMKD (Zhang et al., 2024b). Besides, to ensure a fair comparison with parameter-based FL works (*e.g.*, FedRCL), in our experiments, we exchange both model parameters and prototypes during training for FedProto, FedTGP, and FedSA (denoted by † in the following tables, please see analysis in Appendix Table A7).

**Configurations.** The global rounds $R$ as 100, clients $K$ as 10, and local epochs $E$ as 10 are used, where all methods have little or no gain with more rounds. We follow (Li et al., 2020b; 2021) and use SGD to update model with a learning rate $\eta$ of 0.01, a momentum of 0.9, a batch size of 64, and a weight decay of $10^{-5}$. The dimension $d$ of the embedding as 512, $\tau$ in Equation (12) as 0.05, $|\mathcal{I}|$ in Equation (7) as 5, $M$ in Equation (9) as 30 by default. All experiments are run on a NVIDIA A5000 GPU. More details for experimental implementations, please refer to Appendix C.1.

### 5.2. Main Results

**Comparison with Baselines.** Table 1 provides comparison results under different FL scenarios (NID1, NID2, $\rho$). We can observe that FedHPro consistently outperforms eight SOTA baselines, confirming the significance of the hyper-

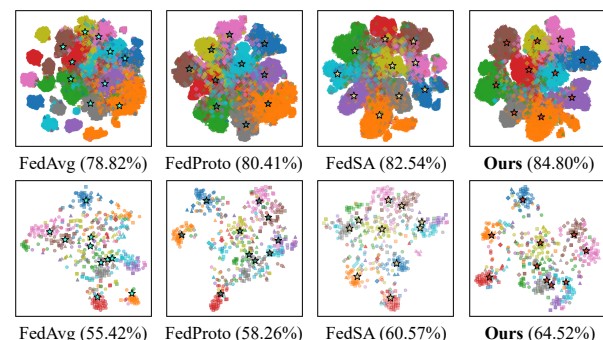

FedAvg (78.82%) FedProto (80.41%) FedSA (82.54%) **Ours** (84.80%)

FedAvg (55.42%) FedProto (58.26%) FedSA (60.57%) **Ours** (64.52%)

*Figure 6.* **T-SNE visualization** on Digits (**Top**) and Office-Caltech (**Bottom**). Each color represents one class, each shape represents one domain, and the stars represent semantic centers.

*Table 3.* **Enabling existing prototype-based methods**, in which their prototypes are replaced with our hyper-prototypes.

| USING HP | DIGITS | | | | |
|---|---|---|---|---|---|
| | MNIST | USPS | SVHN | SYN | AVG↑ |
| FedProto† | 97.48 ↑0.08 | 92.24 ↑0.35 | 80.90 ↑1.68 | 56.39 ↑3.24 | 81.75 ↑1.34 |
| FedTGP† | 97.83 ↑0.10 | 92.67 ↑0.24 | 82.49 ↑1.14 | 57.46 ↑2.27 | 82.61 ↑0.94 |
| FedGMKD | 97.85 ↑0.05 | 92.71 ↑0.45 | 82.52 ↑1.39 | 58.04 ↑1.87 | 82.79 ↑0.96 |
| FedSA† | 97.98 ↑0.20 | 92.79 ↑0.29 | 82.93 ↑1.16 | 59.30 ↑1.18 | 83.25 ↑0.71 |

prototypes in capturing consistent semantic representations and their effectiveness when utilized via HPCL and HPAL. Even with extremely imbalanced classes (*e.g.*, on CIFAR10-LT ($\rho = 100$) where class 0 has 5000 samples while class 9 only has 50 samples), FedHPro still attains optimal results owing to the hyper-prototypes globally learned from real samples through gradient matching. Table 2 further presents the results on Office-Caltech under the domain skew; such a challenging scenario demonstrates FedHPro's ability to promote semantic consistency across domains.

**T-SNE Visualization.** Figure 6 depicts t-SNE (Maaten & Hinton, 2008) visualization of the representation space (*i.e.*, the embedding space after the feature extractor) for different methods on Digits and Office-Caltech. The results show that

*Table 4.* **Efficiency comparison** per round (20 clients) on Digits.

| Methods | Upload$_{\times 10^7}$ | Comm. | FLOPs$_{\times 10^{10}}$ | Time$_{train}$ |
|---|---|---|---|---|
| FedAvg | 10.87 | 380.42MB | 1.02 | 23.11s |
| FedProto[†] | 10.90 | 387.57MB | 1.04 | 25.07s |
| FedTGP[†] | 10.90 | 387.57MB | 1.89 | 45.49s |
| FedGMKD | 10.92 | 392.32MB | 2.06 | 52.31s |
| FedRCL | 10.87 | 380.42MB | 1.73 | 41.69s |
| FedSA[†] | 10.90 | 387.57MB | 1.37 | 36.21s |
| FedHPro | 10.90 | 387.57MB | 1.35 | 35.16s |

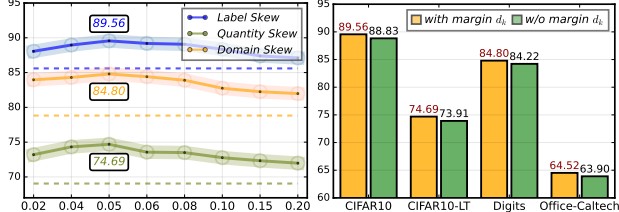

*Figure 7.* Settings: label skew (CIFAR10 with NID1$_{0.5}$), quantity skew (CIFAR10-LT with NID1$_{0.5}$, $\rho = 50$), domain skew (Digits). **Left**: different $\tau$ of Equation (12), **dotted lines** denote the FedAvg. **Right**: with or without (*w/o*) the margin $d_k$ of Equation (12).

FedHPro yields a larger inter-class distance (*i.e.*, different colors are more separated) and a reduced intra-class distance (*i.e.*, different shapes of the same color are clustered more compactly). This indicates FedHPro's success in establishing a more generalizable decision boundary across diverse domains under challenging heterogeneous FL scenarios.

**Hyper-Prototypes as a Plug-and-Play Component.** As previously shown in Figure 2, hyper-prototypes can be easily incorporated with FedProto to improve model performance. Table 3 reports the accuracy of four existing prototype-based FL methods with their prototypes replaced with our hyper-prototypes on the Digits. The hyper-prototypes consistently improve the accuracy of these methods in multiple domains, particularly the SYN domain. This further confirms that our hyper-prototypes preserve better semantic consistency than the conventional prototypes during FL training.

**Efficiency Comparison.** Table 4 compares communication and computation costs on the Digits. FedHPro acquires satisfactory performance with moderate training cost. Compared to FedAvg, we only additionally transmit the average gradient. Besides, FedHPro achieves competitive computation costs (*e.g.*, FLOPs) against FedSA, where our training costs mainly come from the contrastive term of HPCL.

**Comparison Results of Convergence Rates.** We plot the average accuracy per epoch for FedHPro and other baselines in Figure 9. Our FedHPro required fewer rounds to converge compared with other baselines and achieved a higher final accuracy, showing the superiority of FedHPro in dealing with non-IID data and its robust stability. These results further indicate that FedHPro helps to promote both

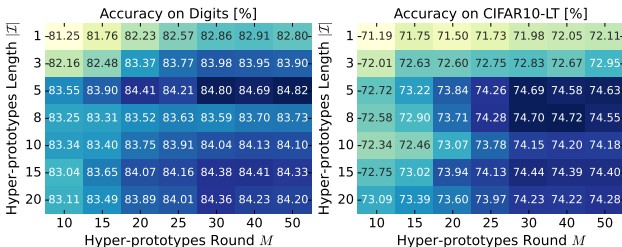

*Figure 8.* **Analysis of hyper-prototypes** lengths $|\mathcal{I}|$ of Equation (7) and rounds $M$ of Equation (9) for Digits under domain skew (**Left**) and CIFAR10-LT under quantity skew (**Right**).

*Table 5.* **Ablation study for two modules** of FedHPro: HPCL and HPAL, on the Digits (**Top**) and Office-Caltech (**Bottom**).

| HPCL | HPAL | MNIST | USPS | SVHN | SYN | AVG ↑ |
|---|---|---|---|---|---|---|
| ✗ | ✗ | 96.68 | 90.43 | 76.35 | 51.85 | 78.82 |
| ✓ | ✗ | 98.06 | 91.74 | 83.81 | 60.74 | 83.58 |
| ✗ | ✓ | 98.38 | 91.93 | 84.16 | 61.30 | 83.94 |
| ✓ | ✓ | 98.52 | 93.13 | 84.95 | 62.59 | **84.80** |
| **HPCL** | **HPAL** | Caltech | Webcam | Amazon | DSLR | AVG ↑ |
| ✗ | ✗ | 60.71 | 48.90 | 75.79 | 36.28 | 55.42 |
| ✓ | ✗ | 63.27 | 57.10 | 79.64 | 43.67 | 60.92 |
| ✗ | ✓ | 64.16 | 55.35 | 79.11 | 46.75 | 61.34 |
| ✓ | ✓ | 64.61 | 62.45 | 80.69 | 50.33 | **64.52** |

the generalizability and efficiency during the FL training. The comparison results of convergence rates on Digits and Office-Caltech datasets are provided in Table A4.

**Impact of Model Heterogeneity.** As shown in Table A9, we report the performance of FedHPro and other baselines on four types of model heterogeneity. We observe that the performance of the prototype-based FL methods significantly decreases as the number of feature extractors X increases. This suggests that prototype-based FL methods heavily rely on the quality of data representations. Both averaging local prototypes (*e.g.*, FedProto) and refining global anchors (*e.g.*, FedSA) lead to semantic drift, resulting in catastrophic performance degradation. In contrast, FedHPro parameterizes semantic anchors as a set of learnable hyper-prototypes, optimized via gradient matching to align with class-relevant characteristics distilled from clients' real samples, thereby effectively mitigating this semantic drift.

### 5.3. Validation Analysis

To thoroughly analyze the validity of essential modules in FedHPro, we perform the following ablative studies:

**Hyper-Parameter Analysis.** We first present a quantitative result to investigate the effect of temperature $\tau$ in Equation (12), as shown in Figure 7-*Left*, which reveals that a smaller $\tau$ is more beneficial than the higher ones, and FedHPro works stably *w.r.t.* $\tau \in [0.04, 0.06]$. Besides, accuracy improves progressively with increasing $\tau$, and the

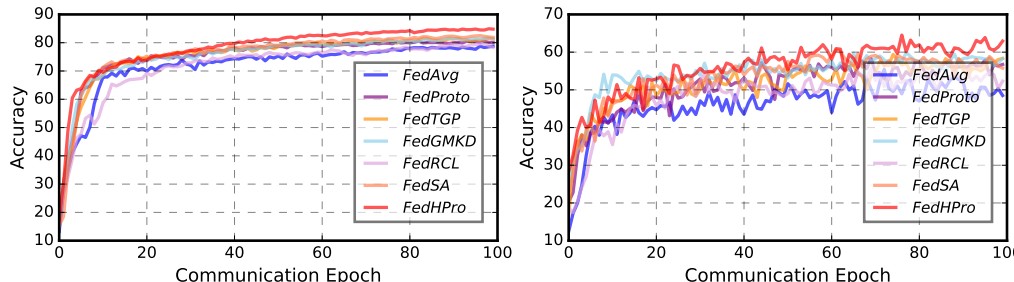

*Figure 9.* Comparison of convergence in the **Average Accuracy Trend** on the Digits (**Left**) and Office-Caltech (**Right**). We also provide quantitative results for the convergence rates in Table A4.

*Table 6.* **Comparison for hyper-prototypes and global prototypes** ($\mathbb{P}$ in Equation (4)), $\Rightarrow$ means replace with $\mathbb{P}$ in the loss.

| USING PROTOTYPES | DIGITS | | OFFICE-CALTECH | | CIFAR10-LT | |
|---|---|---|---|---|---|---|
| | AVG ↑ | △ | AVG ↑ | △ | ACC ↑ | △ |
| Hyper-Prototypes | **84.80** | – | **64.52** | – | **74.69** | – |
| Using $\mathbb{P} \Rightarrow$ HPCL | 83.72 | -1.08 | 63.15 | -1.37 | 73.22 | -1.47 |
| Using $\mathbb{P} \Rightarrow$ HPAL | 83.29 | -1.51 | 62.42 | -2.10 | 72.89 | -1.80 |
| Using $\mathbb{P} \Rightarrow$ BOTH | 81.35 | -3.45 | 59.69 | -4.83 | 72.56 | -2.13 |

amelioration becomes negligible when $\tau = 0.05$. Then, as shown in Figure 7-*Right*, we analyze the effect of $d_k$ in Equation (12). The results confirm that the client-specific margin $d_k$ encourages better class-wise representations by adaptively sharpening the decision boundary, yielding consistent improvements across different FL scenarios.

We next focus on the hyper-prototypes $\mathcal{S}_M$ and provide the overall performance with different lengths $|\mathcal{I}|$ (Equation (7)) and rounds $M$ (Equation (9)). As presented in Figure 8, we observe that the accuracy improves gradually as $|\mathcal{I}|$ and $M$ increase, but an excessively high $|\mathcal{I}|$ is detrimental to training due to optimization instability. Moreover, enlarging $M$ leads to higher accuracy in FedHPro, and the gain becomes marginal when $M = 30$. Empirically, to achieve a trade-off on efficiency, $|\mathcal{I}| = 5$ and $M = 30$ are recommended. The ablation studies of the number of clients $K$, local epochs $E$, and feature dimension $d$ are provided in Appendix C.2.

**Module Ablation.** We first provide an empirical result in Table 5 for the two modules of FedHPro (the first row refers to FedAvg). We observe that: 1) HPCL brings significant gains over the baseline, indicating that our HPCL is able to promote inter-class separability while preserving semantics; 2) HPAL also yields solid improvements, proving the importance of aligning embeddings across clients via smooth regularization; and 3) Combining HPCL and HPAL attains optimal accuracy, indicating they complement each other in enhancing FL local training under data heterogeneity.

To further confirm the effectiveness of hyper-prototypes, as shown in Table 6, we use global prototypes $\mathbb{P}$ (Equation (4)) to replace our hyper-prototypes in HPCL (Equation (12)) or HPAL (Equation (13)). It is evident that using global proto-

types largely weakens both HPCL and HPAL, indicating the superiority of the hyper-prototypes over the conventional prototypes from the proposed module perspective.

**Extra Results.** The analysis of clients $K$ and local epochs $E$ is provided in Table A5. As shown in Table A6, we also investigate the feature dimension $d$ in FedHPro to evaluate its impact on model performance. Table A8 demonstrates that FedHPro is easier to obtain a fair solution than FedAvg. Besides, Table A9 further illustrates the effectiveness of FedHPro under model heterogeneity. Table A10 also shows the applicability of FedHPro to the text-modality scenario. More details, please see Appendix Section C.2.

## 6. Conclusion

This paper proposes hyper-prototypes, which are optimized via gradient matching to align with class-relevant characteristics distilled from real samples rather than prototype-level descriptors, promoting substantially improved semantic consistency across clients. By leveraging hyper-prototypes, we further propose FedHPro with two key modules, HPCL and HPAL, to encourage the inter-class separability and intra-class uniformity during FL local training. Extensive experiments on diverse FL scenarios validate the effectiveness of FedHPro and the contribution of hyper-prototypes.

## Acknowledgment

This work is supported by A*STAR under its MTC YIRG Grant (M24N8c0103), the Lee Kong Chian Fellowship (T050273), and the Singapore Ministry of Education (MOE) Academic Research Fund (AcRF) Tier 1 Grant (24-SIS-SMU-008).

## Impact Statement

This work advances prototype-based FL to better handle heterogeneous clients while keeping training data decentralized, which could enable more reliable collaboration in privacy-sensitive settings (*e.g.*, financial monitoring).

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

# A. Algorithm Pseudo-code Flow

In this section, we describe the pseudo-code of our FedHPro in Algorithm 1: 1) *Server-Side*: we optimize the simulated hyper-prototypes via gradient matching; 2) *Client-Side*: we leverage the hyper-prototypes to promote FL local training.

---

**Algorithm 1**  FedHPro: Federated Hyper-Prototype Learning

---

**Input:** Communication Rounds $R$, Local Epochs $E$, Clients $K$, $k$-th client's data $\mathcal{D}_k$ and model $w_k$
**Output:** The final FL global model $w_R^*$
`/*` `Server-Side Executing` `*/`
**for** each communication round $r = 1, 2, ..., R$ **do**
  $\mathcal{A}_r \Leftarrow \{1, ..., K\}$ `// randomly select clients`
  **for** each client $k \in \mathcal{A}_r$ **in parallel do**
    $w_k, \mathbf{g}_k \leftarrow$ **LocalUpdating**$(w_r^*, \mathcal{S}_M)$
  **end**
  $w_{r+1}^* \leftarrow \sum_{k=1}^{|\mathcal{A}_r|} \frac{n_k}{N} w_k$ `// global aggregation`
  `/* optimizing hyper-prototypes (Section 2.2) */`
  $\mathcal{G} = [\mathbf{g}^1, \ldots, \mathbf{g}^c, \ldots, \mathbf{g}^{\mathbb{C}}] \leftarrow$ Equation (6)
  $\mathcal{G}_{\text{HP}} = [\mathbf{g}_{\text{HP}}^1, \ldots, \mathbf{g}_{\text{HP}}^c, \ldots, \mathbf{g}_{\text{HP}}^{\mathbb{C}}] \leftarrow$ Equation (7)
  $\mathcal{L}_{GM}(\mathcal{G}, \mathcal{G}_{\text{HP}}) \leftarrow$ Equation (8) `// gradient matching`
  $\mathcal{S}_M = [\mathcal{S}_M^1, \ldots, \mathcal{S}_M^c, \ldots, \mathcal{S}_M^{\mathbb{C}}] \leftarrow$ Equation (9)
**end**
`/*` `Client-Side Updating` `*/`
**LocalUpdating**$(w_r^*, \mathcal{S}_M)$:
$w_k \leftarrow w_r^*$ `// distribute global parameters`
**for** each local epoch $e = 1, 2, ..., E$ **do**
  **for** each batch $b \in$ private data $\mathcal{D}_k$ **do**
    $\mathcal{L}_{CE} = \sum_{i \in b} -\mathbf{1}_{y_i} \log(\delta(h_k(z_i))) \leftarrow$ Equation (14)
    `/* Hyper-Prototype Contrastive Learning (Section 3.1) */`
    $d_k \leftarrow$ Equation (10) `// client margin`
    $\mathcal{L}_{HPCL}(z_i, \mathcal{S}_M^c, \mathcal{N}_M^c, d_k)_{i \in b} \leftarrow$ Equation (12)
    `/* Hyper-Prototype Alignment Learning (Section 3.2) */`
    $\mathcal{L}_{HPAL}(z_i, \mathcal{H}_M^c)_{i \in b} \leftarrow$ Equation (13)
    $\mathcal{L} = \mathcal{L}_{CE} + \mathcal{L}_{HPCL} + \mathcal{L}_{HPAL}$
    $w_k \leftarrow w_k - \eta \nabla \mathcal{L}(w_k; b)$ `// update model`
  **end**
**end**
$\mathbf{g}_k = \{\}$ `// initialize gradients`
**for** each class $c = 1, 2, ..., \mathbb{C}$ **do**
  $\mathbf{g}_k^c = \frac{1}{n_k^c} \sum_{(x_i, y_i) \in \mathcal{D}_k^c} \nabla_{z_i} \mathcal{L}_k(x_i, y_i) \leftarrow$ Equation (5)
  $\mathbf{g}_k = \mathbf{g}_k \cup \{\mathbf{g}_k^c\}$
**end**
return $w_k, \mathbf{g}_k$ to the server

---

# B. Convergence Analysis of FedHPro

In this section, we provide a theoretical analysis of the proposed FedHPro under non-convex objectives. This analysis of FedHPro is grounded in four standard assumptions prevalent in the field of FL, which are similar to existing general works (Li et al., 2020b; Ye et al., 2023; Gao et al., 2022; Li et al., 2020d). We describe the notations in Section B.1, the standard assumptions in Section B.2, and the proofs in Section B.3.

## B.1. Preliminaries

We introduce some additional variables to better represent the FL training process and model updates from a mathematical perspective. For the $k$-th client, based on Section 2.1, we define local model's parameters as $w_k = \{\phi_k, \mu_k\}$, where $f_k(\phi_k)$ is a feature extractor maps each sample $x$ to a $d$-dim embedding $z = f_k(x)$; and $h_k(\mu_k)$ is a classifier maps the embedding $z$ into a $\mathbb{C}$-dim output vector $\ell = h_k(z)$. In addition, $\mathcal{D}_k = \{x_i, y_i\}|_{i=1}^{n_k}$ is the training dataset for $k$-th client, $n_k$ is the number of total samples, the label $y_i$ belongs to one of $\mathbb{C}$ categories and is expressed as $y_i \in \{1, 2, \ldots, \mathbb{C}\}$. In each communication round, each client $k$ uploads their local model parameters $w_k$ and average gradients $\mathbf{g}_k = \{\mathbf{g}_k^1, \ldots, \mathbf{g}_k^c, \ldots, \mathbf{g}_k^{\mathbb{C}}\}$ to the

server. Then, the server optimizes hyper-prototypes via gradient matching (Section 2.2) and aggregates local models to get the global model. In the next round, both the global model and hyper-prototypes are broadcast to each participant. We denote $\mathcal{L} = \mathcal{L}_{CE} + \mathcal{L}_{HPCL} + \mathcal{L}_{HPAL}$ as $\mathcal{L}_r$ with a subscript indicating the round $r$ of global rounds.

Besides, we express $\mathbf{e} \in \{1, 2, \ldots, E\}$ as the local iteration step, $r$ as the global round, $E$ as the local epochs, $rE + \mathbf{e}$ refers to the $\mathbf{e}$-th local update in the round $r + 1$, and $rE + 1$ denotes the time between $r$-th global aggregation at the server and starting the model update on round $r + 1$ at the local client. Therefore, if the step is at $rE + 1$ of the local iterations, the local model is the same as the step at $rE$ of the local iterations.

## B.2. Assumptions

Let $w_{k,r} = \{\phi_{k,r}, \mu_{k,r}\}$ as the local model's parameters of $k$-th client at round $r$, $f_k(\phi_k)$ as a feature extractor, and $h_k(\mu_k)$ as a classifier. Our convergence analysis is based on the following four standard assumptions in FL:

**Assumption B.1** (Smoothness). Each objective function $\mathcal{L}$ is $L_1$-Lipschitz smooth, which suggests the gradient of objective function $\mathcal{L}$ is $L_1$-Lipschitz continuous:

$$\|\nabla \mathcal{L}_{k,r_1} - \nabla \mathcal{L}_{k,r_2}\|_2 \leq L_1 \|w_{k,r_1} - w_{k,r_2}\|_2, \quad \forall \, r_1, r_2 > 0, \;\; k \in \{1, \ldots, K\}. \tag{A1}$$

Based on the derivative of the differentiable function, it further implies the following quadratic bound ($L_1 > 0$) as:

$$\mathcal{L}_{k,r_1} - \mathcal{L}_{k,r_2} \leq (\nabla \mathcal{L}_{k,r_2})^\top (w_{k,r_1} - w_{k,r_2}) + \frac{L_1}{2} \|w_{k,r_1} - w_{k,r_2}\|_2^2. \tag{A2}$$

**Assumption B.2** (Unbiased Gradient and Bounded Variance). For each client, the stochastic gradient is unbiased ($\xi$ is the sample of the client's dataset), so the unbiased gradient is expressed as follows:

$$\mathbb{E}_\xi [\nabla \mathcal{L}(w_{k,r}, \xi)] = \nabla \mathcal{L}(w_{k,r}) = \nabla \mathcal{L}_r, \quad \forall \, k \in \{1, \ldots, K\}, \;\; r > 0, \tag{A3}$$

and its variance is further bounded by $\sigma^2$ as:

$$\mathbb{E}_\xi [\|\nabla \mathcal{L}(w_{k,r}, \xi) - \nabla \mathcal{L}(w_{k,r})\|_2^2] \leq \sigma^2, \quad \forall \, k \in \{1, \ldots, K\}, \;\; \sigma^2 \geq 0. \tag{A4}$$

**Assumption B.3** (Bounded Dissimilarity of Stochastic Gradients). For each objective function $\mathcal{L}$, there exists a constant $B > 0$ such that the stochastic gradient is bounded:

$$\mathbb{E}_\xi [\|\nabla \mathcal{L}(w_{k,r}, \xi)\|_2] \leq B, \quad \forall \, k \in \{1, \ldots, K\}, \;\; B > 0, \;\; r > 0. \tag{A5}$$

**Assumption B.4** (Continuity). For each real-valued function $f(\phi)$ is $L_2$-Lipschitz continuous:

$$\|f_{k,r_1}(\phi_{k,r_1}) - f_{k,r_2}(\phi_{k,r_2})\| \leq L_2 \|\phi_{k,r_1} - \phi_{k,r_2}\|_2, \quad \forall \, r_1, r_2 > 0, \;\; k \in \{1, \ldots, K\}. \tag{A6}$$

## B.3. Completing Proofs

In this subsection, based on the above four assumptions, we present the following theorems and prove them:

**Theorem B.1** (Deviation bound of the objective function $\mathcal{L}$). *Let the assumptions hold, after every communication round, the objective function $\mathcal{L}$ of an arbitrary client will be bounded,*

$$\mathbb{E}[\mathcal{L}_{(r+1)E}] \leq \mathcal{L}_{rE} - \left(\eta - \frac{L_1 \eta^2}{2}\right) E B^2 + \frac{L_1 E \eta^2}{2} \sigma^2 + L_2 E \eta B (\mathbb{C} - 1), \tag{A7}$$

where $\eta$ is the learning rate, $\mathbb{C}$ is the number of classes, $\{L_1, L_2, B, \sigma\}$ are constants in the above assumptions (Section B.2). Theorem B.1 gives the deviation bound of $\mathcal{L}_r$, and the convergence of FedHPro can be guaranteed by choosing an appropriate learning rate $\eta$ during the training process.

**Theorem B.2** (Non-convex convergence for FedHPro). *Let the assumptions hold, the objective function $\mathcal{L}$ of an arbitrary client decreases monotonically as the round increases, with the following condition,*

$$\eta_{\mathbf{e}} < \frac{2B^2 - 2L_2 B (\mathbb{C} - 1)}{L_1 (\sigma^2 + B^2)}, \quad \mathbf{e} \in \{1, \ldots, E - 1\}. \tag{A8}$$

**Theorem B.3** (Non-convex convergence rate for FedHPro). *Let the assumptions hold, the round $r$ from $0$ to $R-1$, given any $\varepsilon > 0$, the objective function $\mathcal{L}$ will converge when,*

$$R > \frac{2}{E\eta} \cdot \frac{\mathcal{L}_0 - \min(\mathcal{L}^*)}{2\varepsilon - L_1\eta(\varepsilon + \sigma^2) - 2L_2 B(\mathbb{C} - 1)}, \tag{A9}$$

*and we can further get the following condition for the learning rate $\eta$ as,*

$$\eta < \frac{2\varepsilon - 2L_2 B(\mathbb{C} - 1)}{L_1(\varepsilon + \sigma^2)}, \tag{A10}$$

where $\min(\mathcal{L}^*)$ denotes the optimal solution of $\mathcal{L}$. Theorem B.3 ensures that the expected $L_2$-norm of the gradients can be confined to any bound $\varepsilon$, and also provides the convergence rate of FedHPro: a smaller $\varepsilon$ requires a larger round $R$, which means that the tighter the bound is, the more communication round $R$ is required to reach in FedHPro.

### B.3.1. PROOF OF THEOREM B.1

*Proof of Theorem B.1.* Theorem B.1 is suitable to each client in FedHPro, so we omit the client's notation $k$ and define the gradient descent as $w_{r+1} = w_r - \eta \nabla \mathcal{L}_r(w_r) = w_r - \eta \theta_r$, then we can naturally get:

$$\mathcal{L}_{rE+2} \overset{(1)}{\leq} \mathcal{L}_{rE+1} + (\nabla \mathcal{L}_{rE+1})^\top (w_{rE+2} - w_{rE+1}) + \frac{L_1}{2}\|w_{rE+2} - w_{rE+1}\|_2^2$$
$$= \mathcal{L}_{rE+1} - \eta(\nabla \mathcal{L}_{rE+1})^\top \theta_{rE+1} + \frac{L_1}{2}\|\eta \theta_{rE+1}\|_2^2, \tag{A11}$$

where $(1)$ from $L_1$-Lipschitz quadratic bound of the above Assumption B.1. We execute the expectation on both sides:

$$\mathbb{E}[\mathcal{L}_{rE+2}] \leq \mathcal{L}_{rE+1} - \eta \mathbb{E}[(\nabla \mathcal{L}_{rE+1})^\top \theta_{rE+1}] + \frac{L_1}{2}\eta^2 \mathbb{E}[\|\theta_{rE+1}\|_2^2]$$
$$\overset{(1)}{=} \mathcal{L}_{rE+1} - \eta \mathbb{E}[\|\nabla \mathcal{L}_{rE+1}\|_2^2] + \frac{L_1}{2}\eta^2 \mathbb{E}[\|\theta_{rE+1}\|_2^2]$$
$$\overset{(2)}{\leq} \mathcal{L}_{rE+1} - \eta \mathbb{E}[\|\nabla \mathcal{L}_{rE+1}\|_2^2] + \frac{L_1}{2}\eta^2 (\mathbb{E}^2[\|\nabla \mathcal{L}_{rE+1}\|_2] + \texttt{Variance}[\|\theta_{rE+1}\|_2])$$
$$= \mathcal{L}_{rE+1} - \eta\|\nabla \mathcal{L}_{rE+1}\|_2^2 + \frac{L_1}{2}\eta^2 (\|\nabla \mathcal{L}_{rE+1}\|_2^2 + \texttt{Variance}[\|\theta_{rE+1}\|_2]) \tag{A12}$$
$$= \mathcal{L}_{rE+1} + (\frac{L_1}{2}\eta^2 - \eta)\|\nabla \mathcal{L}_{rE+1}\|_2^2 + \frac{L_1}{2}\eta^2 \cdot \texttt{Variance}[\|\theta_{rE+1}\|_2]$$
$$\overset{(3)}{\leq} \mathcal{L}_{rE+1} + (\frac{L_1}{2}\eta^2 - \eta)\|\nabla \mathcal{L}_{rE+1}\|_2^2 + \frac{L_1}{2}\eta^2 \sigma^2,$$

where $(1)$ from the stochastic gradient is unbiased in Equation (A3); $(2)$ from the formula: $\texttt{Variance}(x) = \mathbb{E}[x^2] - \mathbb{E}^2[x]$; and $(3)$ from Equation (A4) of Assumption B.2. The $rE + 1$ means the time between the server and client, and $rE + 2$ means the first step. On this basis, we telescope $E$ steps on both sides of the above equation as:

$$\mathbb{E}[\mathcal{L}_{(r+1)E}] \leq \mathcal{L}_{rE+1} + (\frac{L_1}{2}\eta^2 - \eta)\sum_{\mathbf{e}=1}^{E-1}\|\nabla \mathcal{L}_{rE+\mathbf{e}}\|_2^2 + \frac{L_1}{2}E\eta^2\sigma^2. \tag{A13}$$

Then, we consider the relationship between $\mathcal{L}_{rE+1}$ and $\mathcal{L}_{rE}$, that is, given any sample $\xi$, based on Equation (A13), we consider the relationship between $\mathcal{L}_{(r+1)E+1}$ and $\mathcal{L}_{(r+1)E}$, expressed as follows:

$$\mathcal{L}_{(r+1)E+1} = (\mathcal{L}_{CE,(r+1)E+1} + \mathcal{L}_{HPCL,(r+1)E+1} + \mathcal{L}_{HPAL,(r+1)E+1}) + \mathcal{L}_{(r+1)E} - \mathcal{L}_{(r+1)E}$$
$$= (\mathcal{L}_{HPCL,(r+1)E+1} - \mathcal{L}_{HPCL,(r+1)E}) + \mathcal{L}_{(r+1)E}$$
$$\overset{(1)}{\leq} \log(\sum_{\mathcal{S}_M^j \in \mathcal{N}_{M,(r+1)E+1}^c} \exp(s(\xi, \mathcal{S}_M^j))) - \log(\sum_{\mathcal{S}_M^j \in \mathcal{N}_{M,(r+1)E}^c} \exp(s(\xi, \mathcal{S}_M^j))) + \mathcal{L}_{(r+1)E}$$
$$\overset{(2)}{\leq} \sum_{\mathcal{S}_M^j \in \mathcal{N}_M^c} (\|f_{k,(r+1)E}(\xi)\|_2 - \|f_{k,rE}(\xi)\|_2) + \mathcal{L}_{(r+1)E}$$

$$\overset{(3)}{\leq} \mathcal{L}_{(r+1)E} + \sum_{\mathcal{S}_M^j \in \mathcal{N}_M^c} (\|f_{k,(r+1)E}(\xi) - f_{k,rE}(\xi)\|_2)$$

$$\overset{(4)}{\leq} \mathcal{L}_{(r+1)E} + (\mathbb{C}-1)\sum_{k=1}^{K}\frac{n_k}{N}(\|f_{k,(r+1)E}(\xi) - f_{k,rE}(\xi)\|_2)$$

$$\overset{(5)}{\leq} \mathcal{L}_{(r+1)E} + L_2(\mathbb{C}-1)\sum_{k=1}^{K}\frac{n_k}{N}\|\phi_{k,(r+1)E} - \phi_{k,rE}\|_2$$

$$\overset{(6)}{\leq} \mathcal{L}_{(r+1)E} + L_2(\mathbb{C}-1)\sum_{k=1}^{K}\frac{n_k}{N}\|w_{k,(r+1)E} - w_{k,rE}\|_2 \quad\quad (A14)$$

$$= \mathcal{L}_{(r+1)E} + L_2(\mathbb{C}-1)\sum_{k=1}^{K}\frac{n_k}{N}\|\eta\sum_{\mathbf{e}=1}^{E-1}\theta_{k,rE+\mathbf{e}}\|_2$$

$$= \mathcal{L}_{(r+1)E} + L_2(\mathbb{C}-1)\eta\sum_{k=1}^{K}\frac{n_k}{N}\|\sum_{\mathbf{e}=1}^{E-1}\theta_{k,rE+\mathbf{e}}\|_2$$

$$\overset{(7)}{\leq} \mathcal{L}_{(r+1)E} + L_2(\mathbb{C}-1)\eta\sum_{k=1}^{K}\frac{n_k}{N}\sum_{\mathbf{e}=1}^{E-1}\|\theta_{k,rE+\mathbf{e}}\|_2,$$

where (1) from the objective function of Equation (12) of Section 3.1; (2) from Equation (11): $s(z, \mathcal{S}_M^j) = \frac{z}{|\mathcal{I}|}(\sum_{|\mathcal{I}|}\frac{1}{\|z\|_2} \cdot \frac{\mathcal{S}_M^j}{\|\mathcal{S}_M^j\|_2}) \leq z$, then we take expectation to get $\mathbb{E}[s(z, \mathcal{S}_M^j)] \leq \|z\|_2$ with $z = f(\xi)$; (3) from the formula as $\|x\|_2 - \|y\|_2 \leq \|x - y\|_2$; (4) details from Section 2.1 and $\xi$ is the sample of the client dataset; (5) from Equation (A6) of Assumption B.4; (6) from the the fact that $\phi$ is a subset of $w = \{\phi, \mu\}$; (7) from the formula as $\|\sum x\|_2 \leq \sum\|x\|_2$.

Subsequently, we perform the expectation on both sides of the above Equation (A14):

$$\mathbb{E}[\mathcal{L}_{(r+1)E+1}] \leq \mathcal{L}_{(r+1)E} + L_2(\mathbb{C}-1)\eta\sum_{k=1}^{K}\frac{n_k}{N}\sum_{\mathbf{e}=1}^{E-1}\mathbb{E}[\|\theta_{k,rE+\mathbf{e}}\|_2]$$

$$\overset{(1)}{\leq} \mathcal{L}_{(r+1)E} + L_2 E\eta B(\mathbb{C}-1), \quad\quad (A15)$$

where (1) from Assumption B.3, and we can easily get:

$$\mathbb{E}[\mathcal{L}_{rE+1}] \leq \mathcal{L}_{rE} + L_2 E\eta B(\mathbb{C}-1). \quad\quad (A16)$$

Based on Equation (A13) and Equation (A16), we can further get:

$$\mathbb{E}[\mathcal{L}_{(r+1)E}] \leq \mathcal{L}_{rE} + L_2 E\eta B(\mathbb{C}-1) + \frac{L_1}{2}E\eta^2\sigma^2 + (\frac{L_1}{2}\eta^2 - \eta)\sum_{\mathbf{e}=1}^{E-1}\|\nabla\mathcal{L}_{rE+\mathbf{e}}\|_2^2$$

$$\overset{(1)}{\leq} \mathcal{L}_{rE} - \left(\eta - \frac{L_1}{2}\eta^2\right)EB^2 + \frac{L_1 E\eta^2}{2}\sigma^2 + L_2 E\eta B(\mathbb{C}-1), \quad\quad (A17)$$

where (1) from Equation (A5) of Assumption B.3.

Theorem B.1 indicates the deviation bound of the objective function for an arbitrary client after each round, and convergence can be guaranteed when there is a certain expected one-round decrease, which can be achieved by choosing an appropriate learning rate $\eta$. We have completed the proof of Theorem B.1. □

### B.3.2. PROOF OF THEOREM B.2

*Proof of Theorem B.2.* Based on Theorem B.1, we can get:

$$\mathbb{E}[\mathcal{L}_{(r+1)E}] - \mathcal{L}_{rE} \leq -\left(\eta - \frac{L_1}{2}\eta^2\right)EB^2 + \frac{L_1 E\eta^2}{2}\sigma^2 + L_2 E\eta B(\mathbb{C}-1). \quad\quad (A18)$$

Then, to ensure the right side of the above Equation (A18) $\leq 0$, *i.e.*, $-(\eta - \frac{L_1}{2}\eta^2)EB^2 + \frac{L_1E\eta^2}{2}\sigma^2 + L_2E\eta B(\mathbb{C}-1) \leq 0$. We can easily get the following condition for the learning rate $\eta$ as:

$$\eta_e < \frac{2B^2 - 2L_2B(\mathbb{C}-1)}{L_1(\sigma^2 + B^2)}, \quad e \in \{1, ..., E-1\}. \tag{A19}$$

Therefore, the convergence of the objective function $\mathcal{L}$ in FedHPro holds as communication round $r$ increases and the above limitation of the learning rate $\eta$ in Equation (A19). We have completed the proof of Theorem B.2. □

### B.3.3. PROOF OF THEOREM B.3

*Proof of Theorem B.3.* Based on Equation (A17), we can get:

$$\mathbb{E}[\mathcal{L}_{(r+1)E}] \leq \mathcal{L}_{rE} + L_2E\eta B(\mathbb{C}-1) + \frac{L_1}{2}E\eta^2\sigma^2 + (\frac{L_1}{2}\eta^2 - \eta)\sum_{e=1}^{E-1}\|\nabla\mathcal{L}_{rE+e}\|_2^2. \tag{A20}$$

Further, we can naturally get the following equation as:

$$\sum_{e=1}^{E-1}\|\nabla\mathcal{L}_{rE+e}\|_2^2 \leq \frac{2L_2E\eta B(\mathbb{C}-1)}{2\eta - L_1\eta^2} + \frac{2(\mathcal{L}_{rE} - \mathbb{E}[\mathcal{L}_{(r+1)E}]) + L_1E\eta^2\sigma^2}{2\eta - L_1\eta^2}. \tag{A21}$$

Let the round $r$ from $0$ to $R-1$, step e from $1$ to $E$, we can easily get the following equation as:

$$\frac{1}{RE}\sum_{r=0}^{R-1}\sum_{e=1}^{E-1}\mathbb{E}[\|\nabla\mathcal{L}_{rE+e}\|_2^2] \leq \frac{2\sum_{r=0}^{R-1}(\mathcal{L}_{rE} - \mathbb{E}[\mathcal{L}_{(r+1)E}])}{RE(2\eta - L_1\eta^2)} + \frac{2L_2\eta B(\mathbb{C}-1)}{2\eta - L_1\eta^2} + \frac{L_1\eta^2\sigma^2}{2\eta - L_1\eta^2}, \tag{A22}$$

given any $\varepsilon > 0$, $\mathcal{L}$ and $R$ receive the following limitation as:

$$\varepsilon > \frac{\frac{2}{RE}\sum_{r=0}^{R-1}(\mathcal{L}_{rE} - \mathbb{E}[\mathcal{L}_{(r+1)E}])}{2\eta - L_1\eta^2} + \frac{2L_2\eta B(\mathbb{C}-1) + L_1\eta^2\sigma^2}{2\eta - L_1\eta^2}. \tag{A23}$$

Based on Equation (A23), we need to ensure that $\varepsilon > 0$, so we can easily get the following condition for $R$ as:

$$R > \underbrace{\frac{2\left[\mathcal{L}_0 - \min(\mathcal{L}^*)\right]}{E\eta\left[2\varepsilon - L_1\eta(\varepsilon + \sigma^2) - 2L_2B(\mathbb{C}-1)\right]}}_{\mathbf{T_1}}, \tag{A24}$$

where $\min(\mathcal{L}^*)$ denotes the optimal solution of the $\mathcal{L}$, *i.e.*, $\sum_{r=0}^{R-1}(\mathcal{L}_{rE} - \mathbb{E}[\mathcal{L}_{(r+1)E}]) = \mathcal{L}_0 - \mathcal{L}_1 + \mathcal{L}_1 - \mathcal{L}_2 + \cdots + \mathcal{L}_{R-1} - \mathcal{L}_R = \mathcal{L}_0 - \mathcal{L}_R \leq \mathcal{L}_0 - \min(\mathcal{L}^*)$.

The denominator of the above Equation (A24) should satisfy $\mathbf{T_1} > 0$, *i.e.*:

$$2\varepsilon - L_1\eta(\varepsilon + \sigma^2) - 2L_2B(\mathbb{C}-1) > 0. \tag{A25}$$

Therefore, we can easily obtain the following condition for the learning rate $\eta$ as:

$$\eta < \frac{2\varepsilon - 2L_2B(\mathbb{C}-1)}{L_1(\varepsilon + \sigma^2)}. \tag{A26}$$

Theorem B.3 ensures that the expected $L_2$-norm of the gradients can be confined to any bound $\varepsilon$, and also provides the convergence rate of FedHPro: *a smaller $\varepsilon$ requires a larger round $R$, which means that the tighter the bound is, the more communication round $R$ is required to reach in FedHPro.* We have completed the proof of Theorem B.3. □

## C. Experimental Details and Extra Results

In this section, we additionally provide more experimental details and results. All experiments are conducted on 64-bit Ubuntu-18.04 system with a NVIDIA RTX A5000 GPU. Note that all reported results are the average of three repeating runs with a standard deviation of random seeds.

### C.1. Experimental Details

**Implementation Details.** We execute $R = 100$ rounds and set the number of clients as $K = 10$. We conduct local training for $E = 10$ epochs in each communication round using the SGD optimizer with a learning rate $\eta$ of 0.01, a momentum of 0.9, a batch size of 64, and a weight decay of $10^{-5}$. We apply data augmentation (Cubuk et al., 2019) for the TinyImageNet dataset. The label skew heterogeneity level of clients is controlled by the standard deviation $\alpha$ of the Dirichlet distribution, and the quantity skew heterogeneity level is controlled by the index ratio $\rho$ between the sample sizes of the most frequent and the least frequent class. The lower $\alpha$ and higher $\rho$ these are, the more heterogeneous the clients are.

Regarding the crucial hyper-parameters of different baselines, we tune these hyper-parameters from a limited candidate set by grid search: the regularization coefficient of FedProx (Li et al., 2020b) $\mu$ is tuned across $\{10^{-4}, 10^{-3}, 10^{-2}, 10^{-1}\}$ and is selected to be $\mu = 10^{-2}$; the regularization coefficient of MOON (Li et al., 2021) $\mu$ is tuned across $\{0.1, 1.0, 3.0, 5.0, 6.0\}$ and is selected to be $\mu = 1.0$; the regularization coefficient of the FedProto (Tan et al., 2022a) $\lambda$ is tuned across $\{1.0, 2.0, 3.0, 4.0\}$ and is selected to be $\lambda = 1.0$; the temperature $\tau$ of FedRCL (Seo et al., 2024) is selected as $\tau = 0.05$, and the loss coefficient $\beta$ is selected to be $\beta = 1.0$; the number of prototype training epochs $S$ in FedTGP (Zhang et al., 2024a) is selected as $S = 100$, the loss ratio $\lambda$ is tuned across $\{0.1, 0.5, 1.0, 2.0\}$ and is selected to be $\lambda = 1.0$; the loss ratio $\lambda$ of FedGMKD (Zhang et al., 2024b) is tuned across $\{0.2, 0.4, 0.6, 0.8, 1.0\}$ and is selected as $\lambda = 0.6$, and the responsibility ratio $\gamma$ is selected to be $\gamma = 0.5$; the loss ratios $\lambda_1, \lambda_2, \lambda_3$ of FedSA (Zhou et al., 2025) are tuned across $\{0.1, 0.2, 0.3, 0.6, 0.8, 0.9, 1.0\}$, and are selected as $\lambda_1 = 0.1, \lambda_2 = 0.9, \lambda_3 = 1.0$ for our run experiments.

**Datasets and Models.** CIFAR10 and CIFAR100 (Krizhevsky et al., 2009) both have $50,000$ training samples and $10,000$ test samples, and each sample is a $32 * 32$ color image, where CIFAR10 has 10 categories, and CIFAR100 has 100 categories. TinyImageNet (Le & Yang, 2015) contains 200 categories and has $100,000$ training samples and $10,000$ testing samples, and each image's size is $64 * 64$. HAM10000 (Tschandl et al., 2018) is used for skin lesion classification and contains $8,912$ training samples and $1,103$ testing samples with 7 categories, and each sample's size is scaled to $224 * 224$. Then, based on (Kaidi et al., 2019), we further build CIFAR10-LT, CIFAR100-LT, and TinyImageNet-LT datasets, as the unbalanced datasets with long-tailed level $\rho$, $\rho = 1$ denotes globally balanced, and $\rho = 100$ as most imbalanced: where category 0 has 5000 samples, while category 9 only has 50 samples. For the domain-skewed FL scenario, we evaluate on two popular multi-domain datasets: Digits (Peng et al., 2019) includes four domains: MNIST (M), USPS (U), SVHN (SV), and SYN (SY) with 10 categories, the digit number from 0 to 9. Office-Caltech (Gong et al., 2012) consists of four domains: Caltech (C), Webcam (W), Amazon (A), and DSLR (D), each domain with 10 overlapping categories.

We use MobileNetV2 (Sandler et al., 2018) for TinyImageNet and TinyImageNet-LT, and ResNet-10 (He et al., 2016) for the other datasets. The feature dimension $d$ is set as 512 by default. Note that all compared methods employ the same network architecture for fair comparison in different heterogeneous FL scenarios.

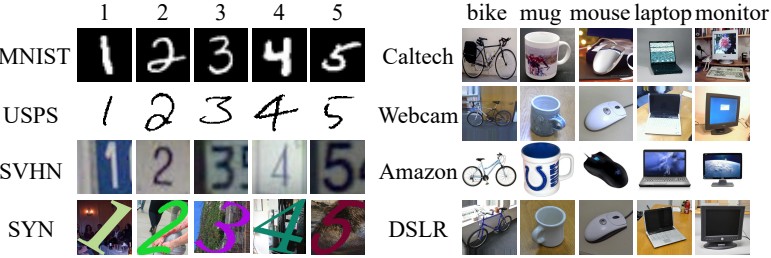

*Figure A1.* Examples of the raw instances from the datasets: Digits (**Left**) and Office-Caltech (**Right**), each has four domains.

**Federated Learning with Data Heterogeneity.** Under heterogeneous FL scenarios, the label distribution $P(y)$, the sample size of different categories, and the conditional feature distribution $P(x|y)$ may vary across participating clients in FL, resulting in different data heterogeneity:

- **Label Skew**: $P_{k_1}(y) \neq P_{k_2}(y)$, $P_{k_1}(x|y) = P_{k_2}(x|y)$, *i.e.*, the label distributions are different among clients, but the features of data samples are similar for each client.

- **Quantity Skew**: A common assumption in quantity skew is that categories are sorted in non-increasing order, *i.e.*, if label $c_1 < c_2$, we have the global sample size $n^{c_1} \geq n^{c_2}$.

- **Domain Skew**: $P_{k_1}(x|y) \neq P_{k_2}(x|y)$, $P_{k_1}(y) = P_{k_2}(y)$, *i.e.*, for the same category space, there are different feature distributions between participating clients.

**Federated Scenarios.** We consider the three heterogeneity settings: 1) **Label Skew**: $\text{NID1}_\alpha$ follows Dirichlet distribution (Wang et al., 2020) ($\alpha$ as the non-IID level), $\text{NID2}$ is a more extreme setting consists of 6 biased clients (each has a single class) and 1 client has all classes. 2) **Quantity Skew**: we shape the original dataset into a long-tailed distribution by (Kaidi et al., 2019), and $\rho$ means the ratio between sample sizes of the most frequent and lowest frequent class. 3) **Domain Skew**: we set 20 and 10 clients and randomly assign these domains for Digits (M: 3, U: 7, SV: 6, SY: 4) and Office-Caltech (C: 3, W: 1, A: 2, D: 4), and each client's dataset is randomly sampled as $1\%$ of the total size for Digits and $20\%$ for Office-Caltech. The domain skew in FL is visualized in Figure A1 on the Digits and Office-Caltech datasets.

**Compared Baselines.** We compare our FedHPro with three types of baselines: 1) Vallina FL with data heterogeneity: FedAvg (McMahan et al., 2017), FedProx (Li et al., 2020b); 2) Contrastive-based FL baselines: MOON (Li et al., 2021), FedRCL (Seo et al., 2024); 3) Prototype-based FL baselines: FedSA (Zhou et al., 2025), FedProto (Tan et al., 2022a), FedTGP (Zhang et al., 2024a), and FedGMKD (Zhang et al., 2024b).

*Prototype-based FL Baselines*: Among these FL methods, FedProto (Tan et al., 2022a), FedTGP (Zhang et al., 2024a), FedGMKD (Zhang et al., 2024b), and FedSA (Zhou et al., 2025) are popular prototype-based FL methods. However, as shown in Table A7, implementing FedProto, FedTGP, and FedSA following their original prototype-only communication setting consistently leads to poor performance. To provide a stronger and fairer reference point, in our experiments, we report *enhanced variants* that transmit prototypes together with model parameters (denoted as FedProto†, FedTGP†, and FedSA† in our tables). We emphasize that these enhanced variants † are not the original algorithms' intended communication protocol: they are upper-bound implementations used solely to decouple effectiveness from communication constraints and to ensure that accuracy comparisons are not dominated by a weaker transmission budget.

## C.2. Additional Experiments

**Comparison Results under Label Skew.** The comparison results for FedHPro and compared FL baselines with different heterogeneity settings ($\text{NID1}_\alpha$ and $\text{NID2}$) are shown in Table A1. We observe that FedHPro universally outperforms other FL baselines, implying that FedHPro can effectively leverage the resulting hyper-prototypes learned from real samples to enhance model training. Experiments show that: 1) FedHPro consistently outperforms others in various non-IID settings, showing that FedHPro is robust to different heterogeneity levels, even in the more difficult $\text{NID2}$ setting; 2) Built upon the HPCL and HPAL modules, FedHPro enhances inter-class separability and enforces intra-class uniformity in local training under heterogeneous FL, yielding improvements over FL baselines.

*Table A1.* Results on CIFAR10, CIFAR100, HAM10000, TinyImageNet with **Label Skew**. † denotes the results obtained by exchanging both the prototypes and model parameters during the FL training process. Best results are in **red bold**, with second best in **blue bold**.

| METHODS | CIFAR10 | | | HAM10000 | | | TINYIMAGENET | | | CIFAR100 | | |
|---|---|---|---|---|---|---|---|---|---|---|---|---|
| | $\text{NID1}_{0.2}$ | $\text{NID1}_{0.5}$ | NID2 | $\text{NID1}_{0.2}$ | $\text{NID1}_{0.5}$ | NID2 | $\text{NID1}_{0.2}$ | $\text{NID1}_{0.5}$ | NID2 | $\text{NID1}_{0.2}$ | $\text{NID1}_{0.5}$ | NID2 |
| FedAvg [AISTATS'17] | 80.59 | 85.60 | 74.48 | 45.26 | 48.68 | 41.54 | 41.34 | 43.88 | 36.20 | 60.08 | 62.48 | 50.56 |
| FedProx [MLSys'20] | 80.74 | 85.54 | 74.67 | 45.42 | 48.55 | 41.21 | 41.16 | 43.50 | 36.62 | 60.21 | 62.53 | 50.29 |
| MOON [CVPR'21] | 82.04 | 86.95 | 76.07 | 47.33 | 50.24 | 43.10 | 43.22 | 44.36 | 37.15 | 61.48 | 63.38 | 51.81 |
| FedProto† [AAAI'22] | 81.61 | 86.25 | 75.32 | 46.35 | 49.70 | 42.45 | 42.25 | 44.19 | 36.86 | 60.26 | 62.91 | 50.60 |
| FedTGP† [AAAI'24] | 84.14 | 87.31 | **77.95** | 48.29 | 50.77 | 44.26 | 43.92 | 45.13 | 38.21 | 62.45 | 64.10 | 52.31 |
| FedGMKD [NeurIPS'24] | 83.56 | **88.09** | 77.29 | 47.97 | 50.49 | 44.20 | 44.09 | 45.30 | 37.94 | 62.76 | 64.62 | 53.05 |
| FedRCL [CVPR'24] | 83.80 | 87.76 | 77.51 | **48.60** | 51.10 | 44.36 | **44.30** | 45.24 | **38.76** | 62.97 | 64.87 | **53.29** |
| FedSA† [AAAI'25] | **84.27** | 87.93 | 77.86 | 48.45 | **51.28** | **44.85** | 44.18 | **45.56** | 38.35 | **63.22** | **65.06** | 52.85 |
| FedHPro (Ours) | **85.98** | **89.56** | **79.70** | **50.23** | **52.79** | **46.17** | **45.64** | **46.90** | **40.52** | **64.41** | **66.25** | **55.08** |

*Table A2.* Results on CIFAR10-LT, CIFAR100-LT, TinyImageNet-LT with **Quantity Skew**. † denotes the results obtained by exchanging both the prototypes and model parameters during the FL training process. Best results are in **red bold**, with second best in **blue bold**.

| METHODS | CIFAR10-LT ($\text{NID1}_{0.5}$) | | | CIFAR100-LT ($\text{NID1}_{0.5}$) | | | TINYIMAGENET-LT ($\text{NID1}_{0.5}$) | | |
|---|---|---|---|---|---|---|---|---|---|
| | $\rho = 10$ | $\rho = 50$ | $\rho = 100$ | $\rho = 10$ | $\rho = 50$ | $\rho = 100$ | $\rho = 10$ | $\rho = 50$ | $\rho = 100$ |
| FedAvg [AISTATS'17] | 75.54 | 69.05 | 60.79 | 53.23 | 45.06 | 34.59 | 34.89 | 25.14 | 21.96 |
| FedProx [MLSys'20] | 75.15 | 69.23 | 60.86 | 52.85 | 44.67 | 34.68 | 34.93 | 25.18 | 21.65 |
| MOON [CVPR'21] | 76.11 | 69.70 | 60.91 | 53.58 | 45.60 | 35.30 | 35.29 | 25.78 | 22.13 |
| FedProto† [AAAI'22] | 75.81 | 69.43 | 60.37 | 53.13 | 45.20 | 34.83 | 34.40 | 25.23 | 21.19 |
| FedTGP† [AAAI'24] | 76.27 | 71.36 | 61.94 | 54.17 | 46.04 | 35.59 | 35.51 | 26.16 | 22.70 |
| FedGMKD [NeurIPS'24] | **76.83** | 71.21 | 62.13 | 54.49 | 46.25 | 35.85 | 35.74 | 26.39 | **23.25** |
| FedRCL [CVPR'24] | 76.60 | 71.55 | 62.41 | 54.43 | **46.73** | **36.27** | 36.10 | 26.54 | 22.91 |
| FedSA† [AAAI'25] | 76.75 | **72.30** | **62.48** | **54.77** | 46.50 | 36.14 | **36.58** | **26.93** | 23.09 |
| FedHPro (Ours) | **78.62** | **74.69** | **64.75** | **55.86** | **47.67** | **37.52** | **37.72** | **28.10** | **24.37** |

*Table A3.* Results on the Digits and Office-Caltech with **Domain Skew**. † denotes the results obtained by exchanging both the prototypes and model parameters during the FL training process. Best results are in **red bold**, with second best in **blue bold**.

| METHODS | DIGITS | | | | | | OFFICE-CALTECH | | | | | |
|---|---|---|---|---|---|---|---|---|---|---|---|---|
| | MNIST | USPS | SVHN | SYN | AVG↑ | △ | Caltech | Webcam | Amazon | DSLR | AVG↑ | △ |
| FedAvg [AISTATS'17] | 96.68 | 90.43 | 76.35 | 51.85 | 78.82 | − | 60.71 | 48.90 | 75.79 | 36.28 | 55.42 | − |
| FedProx [MLSys'20] | 96.61 | 89.92 | 76.80 | 52.69 | 79.01 | +0.19 | 60.77 | 50.83 | 77.27 | 37.61 | 56.62 | +1.20 |
| MOON [CVPR'21] | 95.90 | 91.15 | 76.64 | 43.29 | 76.74 | -2.08 | 57.75 | 46.08 | 72.83 | 34.50 | 52.79 | -2.63 |
| FedProto† [AAAI'22] | 97.40 | 91.89 | 79.22 | 53.15 | 80.41 | +1.59 | 61.58 | 53.21 | 78.72 | 39.53 | 58.26 | +2.84 |
| FedTGP† [AAAI'24] | 97.73 | 92.43 | 81.35 | 55.19 | 81.67 | +2.85 | 61.25 | 54.97 | 78.79 | 42.57 | 59.39 | +3.97 |
| FedGMKD [NeurIPS'24] | 97.80 | 92.26 | 81.13 | 56.17 | 81.83 | +3.01 | 61.78 | 55.36 | 79.30 | 43.22 | 59.91 | +4.49 |
| FedRCL [CVPR'24] | 96.71 | 91.32 | 77.45 | 53.20 | 79.67 | +0.85 | 60.29 | 50.41 | 74.54 | 37.46 | 55.68 | +0.26 |
| FedSA† [AAAI'25] | 97.78 | 92.50 | 81.77 | 58.12 | **82.54** | **+3.72** | 62.23 | 55.85 | 79.13 | 45.05 | **60.57** | **+5.15** |
| FedHPro (Ours) | 98.52 | 93.13 | 84.95 | 62.59 | **84.80** | **+5.98** | 64.61 | 62.45 | 80.69 | 50.33 | **64.52** | **+9.10** |

*Table A4.* Results of **Convergence Rates** on Digits and Office-Caltech, where $R_{acc}$ columns denote the minimum number of rounds required to reach *acc* of the test accuracy, AVG columns show the final test accuracy, and $\triangle_{acc}$ represents the speedup ratio of rounds when FedAvg reaches *acc*.

| METHODS | DIGITS | | | | | | OFFICE-CALTECH | | | | | |
|---|---|---|---|---|---|---|---|---|---|---|---|---|
| | $R_{50\%}$ | $R_{60\%}$ | $R_{70\%}$ | $R_{75\%}$ | AVG↑ | $\triangle_{75\%}$↑ | $R_{20\%}$ | $R_{30\%}$ | $R_{40\%}$ | $R_{50\%}$ | AVG↑ | $\triangle_{50\%}$↑ |
| FedAvg [AISTATS'17] | 8 | 10 | 16 | 27 | 78.82 | − | 3 | 5 | 7 | 25 | 55.42 | − |
| FedProx [MLSys'20] | 8 | 10 | 15 | 30 | 79.01 | ↓11.1% | 3 | 6 | 8 | 26 | 56.62 | ↓4.0% |
| MOON [CVPR'21] | 10 | 12 | 19 | 46 | 76.74 | ↓70.3% | 3 | 6 | 13 | 47 | 52.79 | ↓88.0% |
| FedProto† [AAAI'22] | 5 | 8 | 13 | 23 | 80.41 | ↑14.8% | 2 | 3 | 7 | 22 | 58.26 | ↑12.0% |
| FedTGP† [AAAI'24] | 5 | 8 | 11 | 18 | 81.67 | **↑33.3%** | 2 | 3 | 6 | 20 | 59.39 | ↑20.0% |
| FedGMKD [NeurIPS'24] | 5 | 7 | 11 | 20 | 81.83 | ↑25.9% | 1 | 3 | 6 | 20 | 59.91 | ↑20.0% |
| FedRCL [CVPR'24] | 8 | 12 | 20 | 35 | 79.67 | ↓29.6% | 4 | 6 | 15 | 37 | 55.68 | ↓48.0% |
| FedSA† [AAAI'25] | 5 | 7 | 11 | 18 | **82.54** | **↑33.3%** | 1 | 2 | 5 | 17 | **60.57** | **↑32.0%** |
| FedHPro (Ours) | 5 | 6 | 9 | 14 | **84.80** | **↑48.1%** | 1 | 2 | 4 | 12 | **64.52** | **↑52.0%** |

**Comparison Results under Quantity Skew.** The comparison results for FedHPro and compared FL baselines with different imbalanced levels $\rho$ are presented in Table A2. We observe that: 1) FedHPro achieves significantly better results on imbalance settings with long-tail distribution, which confirms the benefit of our hyper-prototypes to regularize local training, even in the scenario of the global-level class imbalance; 2) FedHPro alleviates the negative impact of the long-tailed distribution by leveraging the semantic guidance of hyper-prototypes learned from the real samples.

**Comparison Results under Domain Skew.** The comparison results of Digits and Office-Caltech datasets with domain skew are shown in Table A3, where different clients are equipped with the data samples from different domains: Digits (MNIST: 3, USPS: 7, SVHN: 6, SYN: 4) and Office-Caltech (Caltech: 3, Webcam: 1, Amazon: 2, DSLR: 4). Each client's dataset is randomly selected from the total samples of the corresponding domain, as 1% for Digits and 20% for

*Table A5.* Ablation study of the number of **Clients** $K$ and **Local Epochs** $E$. **Left**: Clients $K$ on TinyImageNet with the label skew $\text{NID1}_{0.5}$. For the experiments with $K = 50$ and $K = 100$ clients, we randomly select 20% of the total clients to participate during the FL training. **Right**: Local Epochs $E$ on the TinyImageNet with the label skew $\text{NID1}_{0.5}$.

| METHODS | $K=10$ | $K=20$ | $K=30$ | $K=50$ | $K=100$ | $E=1$ | $E=5$ | $E=10$ | $E=15$ | $E=20$ |
|---|---|---|---|---|---|---|---|---|---|---|
| FedAvg [AISTATS'17] | 43.88 | 41.31 | 39.06 | 36.21 | 33.79 | 36.92 | 42.96 | 43.88 | 42.79 | 40.38 |
| FedProto$^{\dagger}$ [AAAI'22] | 44.19 | 41.75 | 39.43 | 36.50 | 33.34 | 36.57 | 42.71 | 44.19 | 43.16 | 40.55 |
| FedTGP$^{\dagger}$ [AAAI'24] | 45.13 | 42.53 | 40.18 | 37.28 | 34.74 | 37.40 | 43.63 | 45.13 | 43.91 | 41.24 |
| FedGMKD [NeurIPS'24] | 45.30 | 42.68 | 40.61 | 37.72 | 34.95 | 37.65 | 43.86 | 45.30 | 44.03 | 41.69 |
| FedRCL [CVPR'24] | 45.24 | **43.70** | 41.02 | **39.08** | **36.12** | 37.80 | 44.16 | 45.24 | **44.60** | **42.67** |
| FedSA$^{\dagger}$ [AAAI'25] | **45.56** | 43.55 | **41.15** | 38.84 | 35.39 | **38.13** | **44.29** | **45.63** | 44.52 | 42.48 |
| FedHPro (Ours) | **46.90** | **44.95** | **42.63** | **41.70** | **38.45** | **40.18** | **45.56** | **46.90** | **45.83** | **44.23** |

*Table A6.* Ablation study for the effect of **Feature Dimension** $d$ on the Digits (**Left**) and Office-Caltech (**Right**).

| METHODS | DIGITS | | | | | OFFICE-CALTECH | | | | |
|---|---|---|---|---|---|---|---|---|---|---|
| | $d=64$ | $d=256$ | $d=512$ | $d=1024$ | AVG ↑ | $d=64$ | $d=256$ | $d=512$ | $d=1024$ | AVG ↑ |
| FedProto$^{\dagger}$ [AAAI'22] | 75.96 | 78.43 | 80.41 | 79.46 | 78.56 | 54.71 | 56.34 | 58.26 | 58.05 | 56.84 |
| FedSA$^{\dagger}$ [AAAI'25] | 77.81 | 81.65 | 82.54 | 81.63 | 80.91 | 57.39 | 59.54 | 60.57 | 59.12 | 59.16 |
| FedHPro (Ours) | 80.28 | 83.37 | 84.80 | 83.15 | **82.90** | 61.87 | 62.47 | 64.52 | 63.10 | **62.99** |

Office-Caltech. The experiments show that FedHPro performs significantly better than other FL baselines across domains, which confirms that resulting hyper-prototypes acquire well-generalizable ability and thus effectively boost performance on different domains. For example, FedHPro outperforms the best counterpart (FedSA) with a gap of 2.26% on the Digits dataset and 3.95% on the Office-Caltech dataset under the domain-skewed FL scenario.

**Ablation Study of Clients $K$ and Local Epochs $E$.** The comparison results of the number of clients $K$ on TinyImageNet under the label skew $\text{NID1}_{0.5}$ are shown in Table A5-*Left*. We can observe that FedHPro brings significant performance gains even under the more challenging settings with more clients. These experiments demonstrate the potential of our FedHPro to be applied to real-world scenarios with massive numbers of clients and random client participation. Moreover, these results further show that the improvements from FedHPro are robust to such uncertainties. Note that for the $K = 50$ and $K = 100$ clients, we randomly select 20% of the total clients to participate in the FL training.

Then, we elaborate on the number of local epochs per communication round. We set the number of local epochs $E$ to be in the set $\{1, 5, 10, 15, 20\}$. The comparison results of local epochs $E$ on TinyImageNet under the label skew $\text{NID1}_{0.5}$ are shown in Table A5-*Right*, in which one observes that with increasing $E$, FedAvg performance first increases and then decreases. This is because when the $E$ is too small, the local training cannot converge properly in each communication round. On the other hand, when the $E$ is too large, the model parameters of local clients might be driven too far from the global optimum. Nevertheless, FedHPro yields consistent improvements over the baselines across different choices of $E$.

**Effect of Feature Dimension $d$.** We further investigate the feature dimension $d$ in FedHPro to evaluate its impact on model performance, as shown in Table A6. We can observe that most prototype-based FL methods show better performance with increasing feature dimensions $d$ from $d = 64$ to $d = 512$, but the model performance degrades with an excessively large feature dimension, *e.g.*, $d = 1024$. It becomes more challenging to train the classifiers with too large a feature dimension. From the Table A6, our FedHPro achieves competitive performance across different choices of the dimension $d$, even in $d = 64$, while FedProto lags by 4.32% on Digits and 7.16% on Office-Caltech.

**Prototype-based Protocols.** FedProto (Tan et al., 2022a), FedTGP (Zhang et al., 2024a), and FedSA (Zhou et al., 2025) are popular prototype-based FL methods. However, as shown in Table A7, implementing FedProto, FedTGP, and FedSA following their original prototype-only communication setting consistently leads to poor performance. To ensure a fair comparison with mainstream FL baselines that exchange model parameters (*e.g.*, FedRCL), in our experiments, we report *enhanced variants* that transmit prototypes together with model parameters (denoted as FedProto$^{\dagger}$, FedTGP$^{\dagger}$, and FedSA$^{\dagger}$ in our tables). We emphasize that these enhanced variants † are not the original algorithms' intended communication protocol: *they are upper-bound implementations used solely to decouple effectiveness from communication constraints and*

*to ensure that accuracy comparisons are not dominated by a weaker transmission budget.*

*Table A7.* Results on the Digits and Office-Caltech under two **Prototype-based Protocols**: prototype-only and joint prototype-and-model parameters (denoted by †). Best results are in **red bold**, with second best in **blue bold**.

| METHODS | DIGITS | | | | | | OFFICE-CALTECH | | | | | |
|---|---|---|---|---|---|---|---|---|---|---|---|---|
| | MNIST | USPS | SVHN | SYN | AVG ↑ | △ | Caltech | Webcam | Amazon | DSLR | AVG ↑ | △ |
| FedAvg [AISTATS'17] | 96.68 | 90.43 | 76.35 | 51.85 | 78.82 | – | 60.71 | 48.90 | 75.79 | 36.28 | 55.42 | – |
| FedProto [AAAI'22] | 93.39 | 87.53 | 70.17 | 45.28 | 74.09 | -4.73 | 57.31 | 46.71 | 72.63 | 28.89 | 51.38 | -4.04 |
| FedTGP [AAAI'24] | 93.72 | 88.07 | 70.56 | 46.25 | 74.65 | -4.17 | 59.09 | 46.74 | 73.33 | 30.67 | 52.46 | -2.96 |
| FedSA [AAAI'25] | 94.19 | 87.39 | 71.30 | 47.14 | 75.01 | -3.81 | 59.87 | 46.70 | 73.96 | 33.27 | 53.45 | -1.97 |
| FedProto† [AAAI'22] | 97.40 | 91.89 | 79.22 | 53.15 | 80.41 | +1.59 | 61.58 | 53.21 | 78.72 | 39.53 | 58.26 | +2.84 |
| FedTGP† [AAAI'24] | 97.73 | 92.43 | 81.35 | 55.19 | 81.67 | +2.85 | 61.25 | 54.97 | 78.79 | 42.57 | 59.39 | +3.97 |
| FedSA† [AAAI'25] | 97.78 | 92.50 | 81.77 | 58.12 | **82.54** | **+3.72** | 62.23 | 55.85 | 79.13 | 45.05 | **60.57** | **+5.15** |
| FedHPro (Ours) | 98.52 | 93.13 | 84.95 | 62.59 | **84.80** | **+5.98** | 64.61 | 62.45 | 80.69 | 50.33 | **64.52** | **+9.10** |

**Fairness across Clients.** Following the fairness metrics in (Li et al., 2020c), we verify the advantage of FedHPro in fairness and report the results on Digits and Office-Caltech. We list the average, the worst $10\%$, and the best $10\%$ of test accuracy over all clients across three runs, as shown in Table A8. We also report the variance of the accuracy distribution across clients (the smaller, the fairer) (Li et al., 2020c). The results demonstrate that FedHPro is easier to obtain a fair solution than FedAvg, because the information conveyed by our hyper-prototypes is more independent of the local data distribution compared to the naive FedAvg scheme.

*Table A8.* **Fairness Evaluation** for the average, the worst $10\%$, the best $10\%$, and the variance of the test accuracy on the Digits (**Left**) across 20 clients, and the Office-Caltech (**Right**) across 10 clients.

| METHODS | DIGITS | | | | OFFICE-CALTECH | | | |
|---|---|---|---|---|---|---|---|---|
| | Worst 10% | Best 10% | Average | Variance ↓ | Worst 10% | Best 10% | Average | Variance ↓ |
| FedAvg [AISTATS'17] | 36.85 | 73.09 | 56.39 | 12.81 | 38.18 | 55.90 | 49.76 | 8.92 |
| FedHPro (Ours) | 43.54 | 78.59 | 62.57 | **8.33** | 44.74 | 63.25 | 54.92 | **5.07** |

**Impact of Model Heterogeneity.** Following (Zhang et al., 2024a; Zhou et al., 2025), we simulate the model heterogeneity among clients by assigning model backbones. Specifically, we denote this setting as $\text{HtFE}_X$, where the $\text{FE}_X$ represents the number $X$ of different feature extractors ($f$ in Section 2.1) used in our experiments. Each client $k$ is assigned a corresponding feature extractor (using $k$ modulo $X$). For example, the $\text{HtFE}_8$ setting includes eight different architectures: 4-layer CNN, GoogleNet (Szegedy et al., 2015), MobileNetV2 (Sandler et al., 2018), ResNet-10, ResNet-18, ResNet-34, ResNet-50, and ResNet-101 (He et al., 2016). To generate the feature representations with an identical feature dimension ($d = 512$ by default), we add an adaptive average pooling layer after each feature extractor. In addition, in our implementation, we only aggregate the same classifier globally across clients, while each client retains its own feature extractor locally, and each client's classifier $h$ is defined as a fully connected layer, *i.e.*, nn.Linear(512, num_classes). In this experimental scenario, † represents transmit prototypes together with the classifier $h$'s parameters: FedProto†, FedTGP†, and FedSA†. For the clients $K$ and local epochs $E$ under model heterogeneity, we set $K = 10$ and $E = 10$.

As shown in Table A9, we report the performance of FedHPro and other baselines on four types of model heterogeneity: 1) $\text{HtFE}_2$: 4-layer CNN and ResNet-18; 2) $\text{HtFE}_3$: ResNet-10, ResNet-18, and ResNet-34; 3) $\text{HtFE}_5$: GoogleNet, ResNet-10, ResNet-18, ResNet-34, and ResNet-50; 4) $\text{HtFE}_8$: 4-layer CNN, GoogleNet, MobileNetV2, ResNet-10, ResNet-18, ResNet-34, ResNet-50, and ResNet-101.

We observe that the performance of the prototype-based FL methods significantly decreases as the number of feature extractors $X$ increases. This suggests that prototype-based FL methods heavily rely on the quality of data representations. Both averaging local prototypes (*e.g.*, FedProto) and refining global anchors (*e.g.*, FedSA) lead to semantic drift, resulting in catastrophic performance degradation. In contrast, FedHPro parameterizes semantic anchors as a set of learnable hyper-prototypes, optimized via gradient matching to align with class-relevant characteristics distilled from clients' real samples, thereby effectively mitigating this semantic drift in heterogeneous FL. Even under the challenging $\text{HtFE}_8$ setting,

*Table A9.* Results of **Model Heterogeneity** on the CIFAR100 dataset with label skew $NID1_{0.5}$ using heterogeneous feature extractors ($HtFE_X$: X means the number of different feature extractors across clients). Best results are in **red bold**, with second best in **blue bold**.

| METHODS | HETEROGENEOUS FEATURE EXTRACTORS HtFE | | | |
|---|---|---|---|---|
| | $HtFE_2$ | $HtFE_3$ | $HtFE_5$ | $HtFE_8$ |
| FedProto[†] [AAAI'22] | $40.28 \pm 0.18$ | $35.54 \pm 0.41$ | $33.40 \pm 0.20$ | $25.96 \pm 0.57$ |
| FedTGP[†] [AAAI'24] | $46.54 \pm 0.11$ | $42.36 \pm 0.21$ | $42.08 \pm 0.24$ | $35.59 \pm 0.41$ |
| FedSA[†] [AAAI'25] | $46.91 \pm 0.31$ | $45.26 \pm 0.33$ | $44.85 \pm 0.19$ | $37.06 \pm 0.22$ |
| FedHPro (Ours) | $48.71 \pm 0.09$ | $47.85 \pm 0.54$ | $47.32 \pm 0.27$ | $42.63 \pm 0.35$ |

*Table A10.* **Left**: Comparison results of FedHPro and baselines on the AG News (Zhang et al., 2015) dataset. **Right**: Comparison results of FedHPro and baselines on the SUN397 (Xiao et al., 2010) dataset.

| SETTINGS | AG NEWS | | | | |
|---|---|---|---|---|---|
| | FedAvg | FedProx | FedSA | FedSA[†] | FedHPro |
| AG-$NID_{K10}$ | 82.09 | 80.43 | 75.13 | **84.03** | **86.78** |
| AG-$NID_{K50}$ | 52.71 | 55.14 | 47.53 | **56.92** | **63.06** |

| SETTINGS | SUN397 | | | | |
|---|---|---|---|---|---|
| | FedAvg | FedRCL | FedSA | FedSA[†] | FedHPro |
| $NID1_{0.2}$ | 68.92 | 70.23 | 61.58 | **70.86** | **73.41** |
| $NID1_{0.5}$ | 70.61 | **73.26** | 65.30 | 72.18 | **75.22** |

FedHPro outperforms FedProto by $16.67\%$, FedTGP by $7.04\%$, and FedSA by $5.57\%$, indicating that the proposed FedHPro is more robust and less affected by the model heterogeneity in FL.

**Applicability to Text-modality.** To verify that our proposed FedHPro can also be applied to the text modality, we explore experiments on a text classification dataset, AG News (Zhang et al., 2015). Specifically, we employ TextCNN (Zhang & Wallace, 2017) with a hidden dimension of 32 for AG News, and train the model via Adam optimizer with a learning rate of 0.01. Referring to (Wang et al., 2024), we consider two non-iid scenarios on the AG News: 1) AG-$NID_{K10}$: we partition the dataset to 10 clients, where $50\%$ clients have samples from 2 classes and the other $50\%$ uniform clients have samples from 4 classes; 2) AG-$NID_{K50}$: we partition the dataset to 50 clients, where $80\%$ clients have samples from 2 classes and the other $20\%$ uniform clients have samples from 4 classes. As shown in Table A10 (**Left**), these results demonstrate the effectiveness of the proposed FedHPro in text-modality scenarios, suggesting that its benefits generalize beyond image-modality tasks.

**Extreme Scenarios with Larger Classes.** Based on Equation (9), with fixed $|\mathcal{I}|$, the gradient-matching cost of FedHPro is approximately linear in the number $\mathbb{C}$ of classes. Empirically, FedHPro remains strong on the CIFAR100 ($\mathbb{C} = 100$), TinyImageNet ($\mathbb{C} = 200$), and their long-tailed variants. In Table A5, FedHPro also performs well at clients $K = 100$. This matches Algorithm 1: in FedHPro, after aggregating client outputs, server-side hyper-prototype updates are independent of the number $K$ of clients. As shown in Table A10 (**Right**), we further evaluate FedHPro on the SUN397 (Xiao et al., 2010) dataset, which involves a larger number of classes $\mathbb{C} = 397$. We also set the number of clients as $K = 100$ with an active ratio of $20\%$. These results support the applicability of FedHPro to larger-scale settings.

