# OpenReview forum: "FedHPro: Federated Hyper-Prototype Learning via Gradient Matching"
_ICML.cc/2026/Conference — ICML 2026 regular_

### Official Review · Reviewer_pJii · 2026-03-08

**Soundness:** 3
**Presentation:** 3
**Significance:** 3
**Originality:** 2
**Overall Recommendation:** 5
**Confidence:** 4

**Summary:**

This paper targets semantic drift and global signal misalignment in heterogeneous prototype-based federated learning, and proposes FedHPro, a novel federated hyper-prototype learning framework. FedHPro aligns learnable global class prototypes with real client sample gradients via gradient matching to distill semantic features and ensure cross-client consistency. It further incorporates Hyper-Prototype Contrastive Learning (HPCL) and Hyper-Prototype Alignment Learning (HPAL) to enhance inter-class separability and intra-class consistency, with local training optimized by cross-entropy loss. Extensive experiments verify that FedHPro outperforms state-of-the-art methods across various heterogeneous settings.

**Compliance With Llm Reviewing Policy:**

Affirmed.

**Final Justification:**

I appreciate the authors' detailed response, and all my concerns have been addressed. I suggest supplementing the standard deviations of the experimental results in the final camera-ready version to improve the rigor of the empirical evaluation. I have raised my score and wish your paper a successful acceptance.

**Key Questions For Authors:**

1. Why does Table 4 only present accuracy results without analyzing the efficiency overhead of performance improvements or the computational cost contribution of hyper-prototypes?

2. How does the semantic consistency of hyper-prototypes vary across different feature extractors in model heterogeneity experiments?

3. What are the essential differences between hyper-prototypes, semantic anchors and trainable global prototypes in terms of optimization objectives and semantic consistency implementation?

4. Could the standard deviation of results in Tables 1-2 be provided since each experiment is repeated three times? This can reflect the stability of the proposed method.

**Limitations:**

1. The paper fails to explicitly state the objective limitations of this work. It is suggested to supplement discussions on issues such as high hyperparameter tuning costs.

2. The Impact Statement is oversimplified. It is recommended to add notes on privacy risks such as gradient inversion attacks on uploaded class-averaged gradients to improve its comprehensiveness.

**Strengths And Weaknesses:**

**Strengths:**
1. The proposed hyper-prototype leverages gradient matching to distill semantic features directly from real client samples, rather than conventional prototype aggregation, effectively boosting cross-client semantic consistency.

2. HPCL and HPAL optimize local feature representations from complementary dimensions (inter-class separability, intra-class consistency), enabling high-quality feature learning in diverse heterogeneous scenarios.

3. The paper provides convergence analysis under non-convex settings, deriving quantitative results on convergence rate and learning rate constraints, which offers solid theoretical support for the framework.

**Weaknesses:**

1. FedHPro involves multiple critical hyperparameters. Although recommended values are provided, scenario-specific tuning is required for different datasets and heterogeneity levels, which raises the cost of practical deployment.

2. Client-uploaded class-averaged gradients pose privacy leakage risks (e.g., gradient inversion attacks), yet no additional privacy-preserving mechanisms are designed for gradient information.

3. For fair comparison, baseline methods (FedProto, FedSA) are enhanced with joint prototype and model parameter transmission, but FedHPro’s performance under low-communication settings (prototype-only transmission) is unvalidated, limiting evaluation in resource-constrained FL scenarios.

4. The paper lacks comparisons with representative federated prototype learning baselines, including FPL[1], FedPLVM[2], MPFT[3], and FedPall[4], weakening the verification of method superiority.

> [1] W Huang, et, al., Rethinking federated learning with domain shift: A prototype view, CVPR 2023.
> [2] L Wang, et, al., Taming cross-domain representation variance in federated prototype learning with heterogeneous data domains, NeurIPS 2024.
> [3] J Zhang, et, al., Enhancing federated domain adaptation with multi-domain prototype-based federated fine-tuning, ICLR 2025.
> [4] Y Zhang, et, al., FedPall: Prototype-based adversarial and collaborative learning for federated learning with feature drift, ICCV 2025.

---

> ### Author Rebuttal · Authors · 2026-03-30
>
> Dear Reviewer pJii, we appreciate your valuable time and constructive feedback, and we hope we addressed your concerns. Please let us know if you'd like any further information. Many thanks!
>
> **[W1] FedHPro involves multiple critical hyperparameters**: FedHPro adds only three method-specific controls: $\tau$ in Eq.(12), $|\mathcal{I}|$ in Eq.(7), $M$ in Eq.(9), besides standard FL settings. We do not retune them across datasets or scenarios; all experiments use $\tau$=0.05, $|\mathcal{I}|$=5, $M$=30 by default. Sec 5.3 also shows stable performance for $\tau \in [0.04,0.06]$, and gradual gains with moderate increases in $|\mathcal{I}|$ and $M$. The recommended setting is a safe choice for most scenarios.
>
> **[W2] The class-averaged gradients pose privacy leakage risks**: FedHPro uploads class-wise average gradients, where the server only uses aggregated statistics (with twice linear operations) without direct access to client data: such a design makes individual data reconstruction almost mathematically intractable [R1,R2]. FedHPro is also compatible with privacy mechanisms such as secure aggregation [R3]: due to space limits, please see Reviewer-g2vh [W3]. We will expand these in Sec 2.3.
>
> [R1] Du, Jiacheng, et al. Sok: On gradient leakage in federated learning. USENIX Security 2025
>
> [R2] Pan, Xudong, et al. Exploring the security boundary of data reconstruction via neuron exclusivity analysis. USENIX Security 2022
>
> [R3] Segal, Aaron, et al. Practical secure aggregation for privacy-preserving machine learning. CCS 2017
>
> **[W3] FedHPro under low-communication is unvalidated**: FedHPro at least transmits classifier parameters because Eq.(7) requires simulating gradient computation. We therefore add experiments on CLIFAR10/CIFAR10-LT for original FedProto/FedTGP/FedSA, their variants $\dagger$, and FedHPro$_{CG}$ that transmits only classifier parameters and gradients. The results show that FedHPro remains competitive in resource-constrained FL scenarios.
> |Method|NID1$_{0.2}$|NID1$_{0.5}$|NID2|$\rho=10$|$\rho=50$|$\rho=100$|
> |---|---|---|---|---|---|---|
> |FedProto|76.58|80.42|66.90|69.49|62.14|51.87|
> |FedTGP|79.18|82.70|70.53|72.55|65.36|56.25|
> |FedSA|78.72|83.11|71.75|73.02|65.73|57.60|
> |FedHPro$_{CG}$|82.77|86.09|73.45|75.70|69.12|59.39|
> |FedProto$\dagger$|81.61|86.25|75.32|75.81|69.43|60.37|
> |FedTGP$\dagger$|84.14|87.31|77.95|76.27|71.36|61.94|
> |FedSA$\dagger$|84.27|87.93|77.86|76.75|72.30|62.48|
> |FedHPro|85.98|89.56|79.70|78.62|74.69|64.75|
>
> **[W4] The paper lacks comparisons with representative baselines**: This is a valuable suggestion! We agree these are highly significant baselines, especially under domain skew. MPFT targets pre-trained model fine-tuning, which differs from our from-scratch setting. We thus add experiments for FPL, FedPLVM, FedPall on Office-Caltech. We will add these results in Sec 5.2 and include a discussion on all these four works in Sec. 4.
> |Method|Caltech|Webcam|Amazon|DSLR|Avg.|
> |---|---|---|---|---|---|
> |FPL|62.37|55.78|77.40|47.93|60.87|
> |FedPLVM|62.95|56.08|75.14|46.34|60.13|
> |FedPall|62.74|59.71|79.60|49.59|62.91|
> |FedHPro|64.61|62.45|80.69|50.33|64.52|
>
> **[Q1] Why does Table 4 not analyze the overhead of hyper-prototypes**: The server optimizes only hyper-prototypes, i.e., $\mathbb{C}\times|\mathcal{I}|\times d$ variables over $M$ iterations in feature space. This cost is modest: on Digits (10-class), this costs $3.07×10^{7}$ FLOPs per round, versus $1.02×10^{10}$ FLOPs for FedAvg training; on TinyImageNet (200-class), $1.22×10^{10}$ versus $1.46×10^{14}$. We will revise the paper to clarify this.
>
> **[Q2] How does the semantic consistency of hyper-prototypes in model heterogeneity**: The hyper-prototypes are backbone-agnostic anchors, not tied to any specific extractor. Empirically (Tab.A9), as the number of heterogeneous backbones increases, competing methods degrade more sharply, while FedHPro remains more robust. This is also consistent with Fig.6, Tab.3, and Tab.6.
>
> **[Q3] What are the essential differences between hyper-prototypes, semantic anchors and trainable prototypes**: The semantic anchors (e.g., FedSA) are predefined class references in a narrow semantic space and updated from global prototypes, so their consistency comes from the projector and global prototypes. The trainable global prototypes (e.g., FedTGP) are server-side prototypes aligned to uploaded local prototypes, so they are largely shaped by local training. Our hyper-prototypes instead are optimized by matching virtual gradients to class-wise gradients distilled from real samples, making them a more instance-grounded mechanism for semantic consistency.
>
> **[Q4] Could the standard deviation in Tables 1-2 be provided**: Due to space limits, Tab.1-2 report only mean accuracy over three seeds. We will add full mean±std results in the revision. As a summary, the average std for FedHPro is ±0.17% on CIFAR10 and ±0.26% on CIFAR10-LT, compared to ±0.15% and ±0.43% for FedSA, and ±0.28% and ±0.30% for FedTGP.

---

> > ### Author Rebuttal · Reviewer_pJii · 2026-04-03
> >
> > I appreciate the authors’ response which has addressed some of my previous concerns. However, I still hold persistent worries regarding the security/safety of the proposed method and the fairness of its comparisons with existing approaches. Given these remaining unresolved issues, I will retain my original score.

---

> > > ### Author Response · Authors · 2026-04-03
> > >
> > > Dear Reviewer pJii, thank you for your follow-up and for carefully reassessing our rebuttal.
> > >
> > > We appreciate your remaining concerns regarding security/safety and fairness, and we would like to clarify them in two points:
> > > - **First**, regarding security/safety, FedHPro requires clients to upload class-wise averaged gradients rather than raw samples or features. We agree that the current version should not be interpreted as providing a formal privacy guarantee against gradient leakage. We would like to clarify that: FedHPro is compatible with standard privacy-enhancing mechanisms such as secure aggregation or DP-based protection, which are complementary to our method and can be incorporated without changing its core mechanism. More importantly, we would like to emphasize that the main contribution of this work is the proposed hyper-prototype learning framework, rather than a new privacy-defense mechanism. Within this scope, class-wise averaged gradients provide a sufficiently abstract training signal for realizing our FedHPro. We will revise the Impact Statement to explicitly incorporate these considerations and more clearly discuss the corresponding limitations.
> > > - **Second**, regarding the fairness of comparisons, our rebuttal has already reported results under both settings: the original low-communication setting (Tab.1) and our joint prototype-and-parameter exchange setting (Tab.2). FedHPro remains competitive in both cases, indicating that its gains are not merely a byproduct of the communication setting.
> > >
> > > **\#Tab.1.** Results under original low-communication setting
> > > |Methods|NID1$_{0.2}$|NID1$_{0.5}$|NID2|$\rho=10$|$\rho=50$|$\rho=100$|
> > > |---|---|---|---|---|---|---|
> > > |FedProto|76.58|80.42|66.90|69.49|62.14|51.87|
> > > |FedTGP|79.18|82.70|70.53|72.55|65.36|56.25|
> > > |FedSA|78.72|83.11|71.75|73.02|65.73|57.60|
> > > |**FedHPro$_{CG}$**|82.77|86.09|73.45|75.70|69.12|59.39|
> > >
> > > **\#Tab.2.** Results under joint prototype-and-parameter exchange setting
> > > |Methods|NID1$_{0.2}$|NID1$_{0.5}$|NID2|$\rho=10$|$\rho=50$|$\rho=100$|
> > > |---|---|---|---|---|---|---|
> > > |FedProto$\dagger$|81.61|86.25|75.32|75.81|69.43|60.37|
> > > |FedTGP$\dagger$|84.14|87.31|77.95|76.27|71.36|61.94|
> > > |FedSA$\dagger$|84.27|87.93|77.86|76.75|72.30|62.48|
> > > |**FedHPro**|85.98|89.56|79.70|78.62|74.69|64.75|
> > >
> > > We hope that our further clarification helps address your remaining concerns, and we would be grateful if you could reconsider the score in light of these clarifications and the additional rebuttal results.
> > >
> > > Thank you again for your time, careful assessment, and constructive feedback.

---

### Official Review · Reviewer_JsEA · 2026-03-12

**Soundness:** 3
**Presentation:** 4
**Significance:** 3
**Originality:** 2
**Overall Recommendation:** 4
**Confidence:** 4

**Summary:**

This paper proposes FedHPro, a federated prototype learning framework aimed at mitigating semantic drift caused by non-IID data in federated learning. The key observation is that existing prototype-based FL methods (e.g., FedProto) obtain global prototypes by averaging local prototypes, which may introduce client-specific biases and lead to misaligned global semantic anchors.

To address this issue, the authors introduce hyper-prototypes, a set of learnable global class-wise prototypes optimized via gradient matching. Instead of directly averaging prototypes, the server matches gradients generated from synthetic hyper-prototypes with the class-wise average gradients uploaded by clients. This mechanism aims to approximate global semantic information without requiring access to raw data.

Based on these hyper-prototypes, the proposed framework further introduces two client-side objectives: Hyper-Prototype Contrastive Learning (HPCL) to improve inter-class separability, and Hyper-Prototype Alignment Learning (HPAL) to encourage intra-class consistency across clients.

The authors evaluate the method on several benchmark datasets (e.g., CIFAR-10/100, TinyImageNet, and Digits) under different heterogeneous FL settings. Experimental results show that the proposed approach consistently improves performance over several federated learning baselines.

**Compliance With Llm Reviewing Policy:**

Affirmed.

**Final Justification:**

The rebuttal and follow-up improve the clarity of the paper, but my overall assessment remains unchanged. In my view, the main concerns are only partially resolved, particularly regarding the conditional nature of the gradient-matching justification and the limited explanatory strength of the convergence/scalability analysis. I therefore keep my original score.

**Key Questions For Authors:**

Q1: The experiments mainly use a relatively small number of clients (e.g., 𝐾=10). In practical federated learning systems, the number of participating clients can be much larger (e.g., hundreds or thousands). Could the authors comment on how the proposed framework scales with respect to the number of clients, particularly in terms of gradient aggregation and hyper-prototype optimization?

Q2: The proposed method requires clients to upload class-wise gradients to the server. When the number of classes is large, the communication cost may scale with O(C×d). Could the authors discuss the communication overhead compared with prototype-based methods, especially in scenarios with large numbers of classes?

Q3: The hyper-prototypes are optimized via gradient matching through an iterative process on the server. Could the authors clarify the computational overhead of this optimization step compared with standard aggregation methods such as FedAvg or prototype averaging? In particular, how does the server-side cost scale with the number of classes and hyper-prototypes?

Q4: The paper claims that gradient matching enables the hyper-prototypes to approximate centralized prototypes learned from all clients’ data. However, gradients depend on both feature representations and classifier parameters. Could the authors clarify under what conditions matching class-wise gradients can reliably approximate centralized prototypes or the underlying feature distributions?

**Limitations:**

The paper does not explicitly discuss limitations or potential societal impacts. It would be helpful if the authors could include a brief discussion on the limitations of the proposed approach, such as scalability with large numbers of clients or classes, and the additional communication/computation overhead introduced by gradient matching.

**Strengths And Weaknesses:**

**Soundness**

Strengths
1. **Comprehensive empirical evaluation**: The proposed method is evaluated on multiple benchmark datasets (e.g., CIFAR-10/100, TinyImageNet, Digits) under several heterogeneous FL scenarios, including label skew, quantity skew, and domain skew. The results show consistent improvements over several prototype-based and contrastive FL baselines.


2. **Empirical evidence supporting the core design**: The t-SNE visualizations and ablation studies provide empirical evidence suggesting that the proposed hyper-prototypes may produce clearer class separability compared with standard prototype averaging.


Weaknesses
1. **Key assumption behind gradient matching is not well justified**:  The central claim of the paper is that gradient matching enables the hyper-prototypes to approximate the centralized prototypes computed using all clients’ samples. However, the paper does not theoretically justify this assumption. Matching class-wise gradients does not necessarily imply matching the underlying feature distributions or class prototypes, since gradients depend on classifier parameters and the loss landscape. As a result, it remains unclear under what conditions gradient matching can reliably approximate centralized prototypes.
2. **Limited theoretical justification**: The convergence result provided in Appendix B appears to follow a standard FL convergence template and does not directly characterize the specific role of hyper-prototypes. In particular, the condition in Equation (16) requires the denominator to remain positive, which may not hold under highly heterogeneous FL settings, potentially making the bound vacuous in practice.


3. **Server-side computational overhead is not clearly analyzed**: The hyper-prototypes are optimized via gradient matching on the server, which involves iterative optimization. The paper does not provide a detailed discussion of the additional server-side computational cost compared with standard aggregation schemes such as FedAvg.

**Presentation**

Strengths
1. **Clear problem motivation and narrative**:  The paper clearly identifies the issue of semantic drift caused by prototype averaging in heterogeneous FL and provides an intuitive motivation for introducing hyper-prototypes via gradient matching.
2. **Well-organized experimental section**:  The experimental design covers multiple heterogeneity settings and includes ablation studies and visualization results that help illustrate the behavior of the proposed method.

Weaknesses
1. **Some implementation details are not fully specified in the main text**:  For example, the relative weighting between the different loss components (e.g., L_{CE}​, L_{HPCL}​, and L_{HPAL}​) is not explicitly described in the main formulation. Clarifying these design choices and providing sensitivity analysis would improve reproducibility.

**Significance**

Strengths
1. **Addresses an important issue in prototype-based federated learning**:  The work targets the semantic misalignment problem caused by prototype averaging under non-IID data, which is a known challenge in heterogeneous federated learning.


2. **Potentially useful idea for server-side prototype optimization**:  The concept of optimizing global prototypes via gradient matching instead of direct averaging may inspire further work on alternative aggregation strategies in federated learning.


Weaknesses
1. **Scalability concerns remain unclear**:  The proposed framework requires clients to upload class-wise gradients and the server to perform gradient matching. When the number of classes or clients becomes large, the communication and computation overhead may increase substantially. A discussion of scalability to larger FL systems would strengthen the paper.

**Originality**

Strengths
1. **Interesting integration of gradient matching with prototype-based FL**:  The paper adapts gradient matching, commonly used in dataset distillation, to learn global hyper-prototypes in federated learning.


2. **A different perspective on prototype aggregation**:  Instead of directly averaging local prototypes, the method formulates prototype learning as an optimization problem on the server.


Weaknesses
1. **Partial methodological overlap with existing ideas**:  While the hyper-prototype mechanism is interesting, the client-side components (HPCL and HPAL) largely build upon existing contrastive learning and alignment-based regularization techniques commonly used in federated learning.

---

> ### Author Rebuttal · Authors · 2026-03-30
>
> Dear Reviewer JsEA, we appreciate your valuable time and constructive feedback, and we hope we addressed your concerns. Please let us know if you'd like any further information. Many thanks!
>
> **[W1,Q4] The assumption behind gradient matching is not well justified**: We would like to clarify that our intended claim is not that hyper-prototypes recover the underlying distribution, but that they approximate centralized prototypes in a first-order optimization sense. FedHPro matches class-wise gradients, which reflect class geometry in feature space under a fixed classifier. For a classifier $h(z)=Wz+b$ with a cross-entropy loss, the gradients of embedding $z$ (class $c$) as $F_c(z)=\nabla_{z}\ell(h(z),c)=W^{\top}(\delta(Wz+b)-e_{c})$, where $e_{c}$ is the one-hot vector. Let $p_c^{ctp}$ be the centralized prototype and $\mathbf{g}^{c}$ be the averaged gradient, then if $F_c$ is $L_c$-Lipschitz around $p_c^{ctp}$ and locally non-degenerate with a constant $\mu_c>0$, for the hyper-prototype $\mathcal{H}_M^c\in\mathbb{R}^d$, we can get: $$\|\|\mathcal{H}\_{M}^{c}-p\_{c}^{ctp}\|\|\le\frac{1}{\mu_c}(\|\|\frac{1}{|\mathcal{I}|}\sum\_{j=1}^{\|\mathcal{I}\|}F_c(\mathbf{s}_j^c)-\mathbf{g}^c\|\|+L\_c\Delta\_c),$$ where $\Delta_c$ is the average distance of class-$c$ embeddings to $p_c^{ctp}$. Thus, gradient matching can approximate centralized prototypes when the classifier is fixed during matching and class features are sufficiently compact. The conditions are standard in prototype-based FL. We will revise the paper to state this more precisely.
>
> **[W2] The convergence result appears not to directly characterize hyper-prototypes**: The effect of hyper-prototypes appears in Eq.(16) through the term $L_2 E\eta B(\mathbb{C}-1)$, which comes from bounding the contrastive interaction in Eq.(12). This reveals a tradeoff: stronger hyper-prototype interaction may help learning but adds an explicit step-size-controlled term. If the denominator $\mathbf{T}_1$ in Eq.(16) is non-positive, the bound becomes non-informative; however, this does not imply FedHPro fails. As in standard non-convex FL analyses such as Scaffold [R1], the result is meaningful under sufficiently mild drift, e.g., with a sufficiently small $\eta$.
>
> [R1] Karimireddy, S. P., et al. Scaffold: Stochastic controlled averaging for federated learning. ICML 2020
>
> **[W3,Q2,Q3] Server-side computational overhead is not clearly analyzed**: The overhead is modest because the server optimizes only hyper-prototypes $\mathcal{S}_M$ via the virtual loss, i.e., $\mathbb{C}\times|\mathcal{I}|\times d$ variables over $M$ iterations. On Digits (10-class), this costs $3.07×10^{7}$ FLOPs per round, versus $1.02×10^{10}$ FLOPs for FedAvg training; on TinyImageNet (200-class), it is $1.22×10^{10}$ versus $1.46×10^{14}$. The communication of FedHPro is $\mathcal{O}(\mathbb{C}d)$ uplink and $\mathcal{O}(\mathbb{C}|\mathcal{I}|d)$ downlink. Since $|\mathcal{I}|=5$ is a small constant, FedHPro has the same asymptotic order as prototype-based FL. In addition to Tab.4, on TinyImageNet, FedHPro uploads gradients and distributes hyper-prototypes as 22.5MB (10 clients), FedProto/FedTGP transports prototypes as 8.9MB, FedSA as 9.1MB. We will revise the paper to clarify this.
>
> **[W4] Some implementation details are not fully specified**: We would like to clarify that there are no extra weighting coefficients among the three losses in Eq.(15): we avoid additional loss-weight tuning and show that FedHPro remains effective with this simple objective. Tab.5 shows the ablation among the loss terms. For hyper-prototypes updating, please see Reviewer-fbGV [W4], we will add all details to Appendix C.1.
>
> **[W5,Q1] Scalability concerns remain unclear**: With fixed $|\mathcal{I}|$, the gradient-matching cost is approximately linear in the number of classes $\mathbb{C}$. Empirically, FedHPro remains strong on CIFAR100 ($\mathbb{C}=100$), TinyImageNet ($\mathbb{C}=200$), and their long-tailed variants. In Tab.A5, FedHPro also performs well at clients $K=100$. This matches Algorithm 1: after aggregating client outputs, server-side hyper-prototype updates are independent of $K$. Below we further add the results on SUN397 [R2] ($\mathbb{C}=397$, $K=100$, 20% active clients), supporting applicability to larger-scale settings.
> |Setting|FedAvg|FedRCL|FedSA|FedSA$\dagger$|FedHPro|
> |---|---|---|---|---|---|
> |NID1$_{0.2}$|68.92|70.23|61.58|70.86|73.41|
> |NID1$_{0.5}$|70.61|73.26|65.30|72.18|75.22|
>
> [R2] Xiao, Jianxiong, et al. Sun database: Large-scale scene recognition from abbey to zoo. CVPR 2010
>
> **[W6] Partial methodological overlap with existing ideas**: We would like to clarify that the main novelty of FedHPro is not merely adding a contrastive or alignment term, but learning hyper-prototypes by gradient matching to capture class-relevant characteristics, and then using them to drive local training. Thus, HPCL and HPAL are actually hyper-prototype-specific instantiations, rather than generic plug-in losses.

---

> > ### Author Rebuttal · Reviewer_JsEA · 2026-04-01
> >
> > Thank you for the detailed rebuttal and additional clarifications. The response helps address several of my concerns, particularly by clarifying the intended claim behind gradient matching, providing more discussion of the convergence terms, and quantifying the server-side overhead and larger-scale results.
> >
> > That said, my overall assessment remains unchanged. The rebuttal improves the clarity of the work, but in my view the main concerns are only partially resolved: the justification for gradient matching remains conditional and local, and the convergence/scalability discussions, while helpful, do not fully eliminate the original questions. Therefore, I keep my original score and continue to view the paper as roughly in the weak-accept range.

---

> > > ### Author Response · Authors · 2026-04-02
> > >
> > > Dear Reviewer JsEA, thank you again for your careful follow-up and for reading the rebuttal in detail. We appreciate that you found the clarification helpful, and we also understand your remaining hesitation. We respond in three points.
> > >
> > > **First**, we would like to clarify again that gradient matching serves as a first-order approximation mechanism for centralized prototypes under standard prototype-based FL conditions: (1) a fixed classifier during matching, and (2) sufficiently compact class-wise embeddings. Importantly, these two conditions are not additional assumptions, but inherent characteristics in prototype-based FL.
> > >
> > > **Second**, in the rebuttal, the theoretical picture is internally consistent: the gradient-matching provides a rationale for the hyper-prototypes, while the convergence theorem characterizes the optimization behavior of the full FedHPro objective. In the rebuttal, we also considered the case where the denominator is negative, which is consistent with standard FL works, such as Scaffold: when the drift is sufficiently mild, e.g., with a sufficiently small $\eta$, the bound is meaningful and the convergence statement applies.
> > >
> > > **Finally**, for scalability, the per-round efficiency table shows that FedHPro stays in essentially the same communication regime as prototype-based baselines, with competitive FLOPs and training time. Thus, while our method does introduce an additional server-side optimization step, the empirical evidence suggests that this overhead is moderate in practice.
> > >
> > > Thank you again for your constructive discussion.
> > >
> > > We hope this helps clarify the contribution and scope of FedHPro.

---

### Official Review · Reviewer_g2vh · 2026-03-12

**Soundness:** 2
**Presentation:** 3
**Significance:** 3
**Originality:** 2
**Overall Recommendation:** 4
**Confidence:** 3

**Summary:**

This paper proposes FedHPro to address the semantic drift problem in existing prototype-based federated learning, which is caused by directly aggregating heterogeneous local representations. The paper proposes to simulate the centralized optimization trajectory using gradient matching to create learnable Hyper-Prototypes on the server. On the client side, it uses Hyper-Prototype Contrastive Learning with an adaptive margin and Hyper-Prototype Alignment Learning for dual alignment. Compared with other baselines, the method achieves better classification accuracy in extreme heterogeneous scenarios, including label skew, long-tail distributions, and cross-domain shifts.

**Compliance With Llm Reviewing Policy:**

Affirmed.

**Final Justification:**

I have read the authors' rebuttal. I suggest that the authors report the results of original FedProto/FedTGP/FedSA (not introducing additional communication overhead) in the camera-ready version if accepted. I'll maintain my score.

**Key Questions For Authors:**

Please refer to weaknesses.

1. Could you explain the experimental setup? Why is the setting more fair?

2. Will the method introduce additional privacy leakage due to the additional information exchange between clients and the server?

**Limitations:**

Yes

**Strengths And Weaknesses:**

Strengths:

1: The proposed Hyper-Prototypes idea is interesting.

2: The method performs well in some scenarios (e.g., long-tail and cross-domain).

3: The paper provides a convergence analysis of the proposed method.

Weaknesses:

1: The experimental setup is strange. For FedProto, FedTGP, and FedSA, both model parameters and prototypes are exchanged. However, why are additional messages exchanged if these methods do not need them?

2. Experiments only focus on vision tasks. More experiments on other types of work will be better.

3. The proposed method introduces additional message exchange, which may introduce additional privacy concerns.

---

> ### Author Rebuttal · Authors · 2026-03-30
>
> Dear Reviewer g2vh, we appreciate your valuable time and constructive feedback, and we hope we addressed your concerns. Please let us know if you'd like any further information. Many thanks!
>
> **[W1] The experimental setup is strange for FedProto, FedTGP, FedSA**: Thank you for your important comment. The additional messages are not part of the original designs of FedProto, FedTGP, and FedSA. In our experiments (please see Tab.A7), these methods are evaluated in two senses: their native prototype-only protocols, and variants (marked by $\dagger$) that exchange both prototypes and model parameters. We would like to clarify that the reason for introducing these $\dagger$ variants is to compare against FedHPro and other parameter-sharing FL baselines under a more comparable transmission budget, rather than letting the performance be bounded by a weaker communication protocol. Additional experiments are added below for original FedProto, FedTGP, and FedSA on CIFAR10/CIFAR10-LT (10 clients, 10 local epochs): our method can achieve greater improvement compared to these methods in their original protocols. We will revise our paper to clearly separate these two settings (prototype-only and joint prototype-and-model parameters) and provide results in the tables.
> |Methods|NID1$_{0.2}$|NID1$_{0.5}$|NID2|$\rho=10$|$\rho=50$|$\rho=100$|
> |---|---|---|---|---|---|---|
> |FedProto|76.58|80.42|66.90|69.49|62.14|51.87|
> |FedProto$\dagger$|81.61|86.25|75.32|75.81|69.43|60.37|
> |FedTGP|79.18|82.70|70.53|72.55|65.36|56.25|
> |FedTGP$\dagger$|84.14|87.31|77.95|76.27|71.36|61.94|
> |FedSA|78.72|83.11|71.75|73.02|65.73|57.60|
> |FedSA$\dagger$|84.27|87.93|77.86|76.75|72.30|62.48|
> |FedHPro|85.98|89.56|79.70|78.62|74.69|64.75|
>
> **[W2] Experiments only focus on vision tasks; more experiments on other types of work will be better**: We agree that evaluating beyond vision would further strengthen our paper. In addition to medical, natural, and artificial datasets, we conduct additional experiments for text-modality on AG News [R1] (a text classification dataset). We use TextCNN [R2] with hidden-dimension 32 for AG News, and train the model via Adam optimizer with a learning rate 0.01. Referring to [R3], we consider two non-iid scenarios on AG News: (1) AG-NID$\_{K10}$: we partition the dataset to 10 clients, where 50\% clients have samples from 2 classes and the other 50\% uniform clients have samples from 4 classes; (2) AG-NID$\_{K50}$: we partition the dataset to 50 clients, where 80\% clients have samples from 2 classes and the other 20\% uniform clients have samples from 4 classes. The results in the table below demonstrate the effectiveness of FedHPro in text-modality scenarios.
> |Settings|FedAvg|FedProx|FedSA|FedSA$\dagger$|FedHPro|
> |---|---|---|---|---|---|
> |AG-NID$_{K10}$|82.09|80.43|75.13|84.03|86.78|
> |AG-NID$_{K50}$|52.71|55.14|47.53|56.92|63.06|
>
> [R1] Zhang, Xiang, et al. Character-level convolutional networks for text classification. NeurIPS 2015
>
> [R2] Zhang, Ye, et al. A sensitivity analysis of (and practitioners' guide to) convolutional neural networks for sentence classification. IJCNLP 2017
>
> [R3] Wang, Rui, et al. Personalized federated learning for text classification with gradient-free prompt tuning. NAACL 2024
>
> **[W3] The proposed method may introduce additional privacy concerns**: Thank you for this thoughtful question. In FedHPro, clients need to transmit class-wise average gradients, which are then aggregated on the server and used to optimize hyper-prototypes via gradient matching. FedHPro therefore does not expose raw samples, and the server-side optimization is performed from statistics (with twice linear operations) without direct access to client data: such a design makes individual data reconstruction almost mathematically intractable [R4,R5]. In addition, FedHPro can be naturally extended in a privacy-preserving manner. Specifically, we can send each client's average gradients to the server by using secure aggregation [R6]. For a simple example, the gradients of clients $k_1, k_2$ as $\mathbf{g}\_{k_1}$ and $\mathbf{g}\_{k_2}$; client $k_1$ adds a random noise vector $\mathbf{I}$ to $\mathbf{g}\_{k_1}$ to get $(\mathbf{g}\_{k_1}+\mathbf{I})$, while client $k_2$ substitutes $\mathbf{I}$ from $\mathbf{g}\_{k_2}$ to get $(\mathbf{g}\_{k_2}-\mathbf{I})$. The two modified gradients are then sent to the server to perform Eq.(6) in the paper, i.e., $(\mathbf{g}\_{k_1}+\mathbf{I}+\mathbf{g}\_{k_2}-\mathbf{I})/2=(\mathbf{g}\_{k_1}+\mathbf{g}\_{k_2})/2$. We will revise our paper to include these discussions in Sec. 2.3.
>
> [R4] Du, Jiacheng, et al. Sok: On gradient leakage in federated learning. USENIX Security 2025
>
> [R5] Pan, Xudong, et al. Exploring the security boundary of data reconstruction via neuron exclusivity analysis. USENIX Security 2022
>
> [R6] Segal, Aaron, et al. Practical secure aggregation for privacy-preserving machine learning. CCS 2017

---

> > ### Author Rebuttal · Reviewer_g2vh · 2026-04-02
> >
> > Thank you for your response. I suggest that the authors compare the original methods and do not introduce additional communication overhead in the experiments. I'll keep my score.

---

> > > ### Author Response · Authors · 2026-04-02
> > >
> > > Dear Reviewer g2vh, thank you again for the follow-up and your valuable suggestions.
> > >
> > > We fully respect your caution. We would like to clarify that, in response to your concerns, we have already added three pieces of evidence in our rebuttal:
> > > - (1) the results for the original FedProto/FedTGP/FedSA, which do not introduce additional communication overhead;
> > > - (2) an additional text-modality experiment beyond vision on the AG News dataset;
> > > - (3) a more explicit discussion of the privacy implication of the additional exchanged information, including how the transmitted class-wise average gradients can be combined with secure aggregation.
> > >
> > > Our hope is simply that the newly added evidence makes the paper stronger and addresses the concerns you raised.
> > >
> > > We would sincerely appreciate it if you could reconsider the score in light of these additional results and analyses.
> > >
> > > Thank you again for your time, careful assessments, and constructive feedback.

---

### Official Review · Reviewer_fbGV · 2026-03-12

**Soundness:** 3
**Presentation:** 3
**Significance:** 3
**Originality:** 3
**Overall Recommendation:** 5
**Confidence:** 4

**Summary:**

This paper proposes **FedHPro**, a prototype-based federated learning method that replaces a single averaged global prototype per class with a set of learnable class-wise **hyper-prototypes** optimized on the server via gradient matching. The local training objective combines standard cross-entropy with two additional terms: **Hyper-Prototype Contrastive Learning (HPCL)** to improve inter-class separability, and **Hyper-Prototype Alignment Learning (HPAL)** to improve intra-class uniformity. The paper reports strong empirical gains across label-skew, quantity-skew, and domain-skew benchmarks, and includes an appendix experiment on model heterogeneity.

**Compliance With Llm Reviewing Policy:**

Affirmed.

**Final Justification:**

The rebuttal addresses several of my main concerns, especially by clarifying the privacy implications of the shared gradients and by improving the fairness of the baseline comparison through additional original-protocol results and stronger experimental evidence. I still think some theoretical aspects could be clarified further, particularly the scope of the convergence analysis, but overall the rebuttal strengthens the paper enough that I am now comfortable supporting acceptance. Accordingly, I am increasing my score.

**Key Questions For Authors:**

Please refer to the weaknesses.

**Limitations:**

The limitations of the proposed method should be discussed in greater depth. In particular, the paper should explicitly discuss:
- the restricted nature of the model-heterogeneity setting,
- the dependence on a shared classifier and matched feature dimension,
- the privacy implications of sharing gradients,
- the possibility of reconstruction or label leakage when class counts are very small,
- the scalability of hyper-prototypes to larger label spaces,
- and the fact that the current convergence analysis does not clearly cover the full algorithm.

**Strengths And Weaknesses:**

## Strengths

- The paper addresses an important problem in federated learning: semantic drift of class prototypes under heterogeneous client distributions.
- The core idea is interesting and nontrivial. Learning global semantic anchors through **trainable hyper-prototypes** is a refreshing and promising direction beyond simple prototype averaging.
- The empirical results are strong on the reported benchmarks, and the method consistently outperforms the listed baselines across several heterogeneity settings.
- The experimental section is broad, with ablations on HPCL/HPAL, temperature, hyper-prototype length, optimization rounds, efficiency, fairness, and an appendix experiment on model heterogeneity.
- The “plug-and-play” experiment that replaces conventional prototypes in prior methods with hyper-prototypes is a useful sanity check and supports the value of the proposed idea.

## Weaknesses

- The **theoretical analysis is not fully convincing**.
  - The convergence section defines the analyzed objective as $L = L_{CE} + L_{HPCL} + L_{HPAL}$.
  - However, the full method also relies on server-side hyper-prototype optimization via gradient matching, and it is not clear that this part is covered by the theorem.
  - As written, the analysis seems closer to a result for the local objective than for the full alternating optimization procedure.

- The **baseline comparison is not fully fair in its current form**.
  - The paper modifies methods such as **FedProto** by combining prototype sharing with model averaging, rather than preserving the original method design.
  - I do not think this is a fair comparison practice. Even if the original version performs worse, that is part of the method’s design tradeoff.
  - Once the protocol is changed in this way, it is no longer really the original baseline and should not be presented as if it were.
  - This is particularly important because the original prototype-based methods have much lower communication overhead, which is not reflected fairly once they are augmented with weight sharing.
  - The communication gap would likely become even larger as the number of classes increases.
  - In addition, sharing both model weights and prototypes may also introduce higher privacy risks than the original baseline design.
  - Overall, I think the authors should remain faithful to the original methodologies of the baselines in all experiments, and report modified variants separately if they wish to include them.

- The **privacy discussion needs to be substantially stronger**.
  - Since the method shares gradients with the server, the paper should provide a much more extensive discussion of how client privacy is preserved.
  - In particular, the authors should address whether data reconstruction or label leakage may be possible from the transmitted gradients, especially when a client has only a very small number of samples for a given class.
  - This concern is especially relevant in extreme label-skew settings, where a class prototype or class-wise gradient may effectively summarize only a handful of points.
  - The privacy risks of the proposed communication protocol should be discussed more explicitly rather than only briefly acknowledged.

- Some **implementation and presentation details should be improved**.
  - Section 4 would read better if it were moved immediately after the introduction, since it contains core method details and would help the reader understand the rest of the paper more easily.
  - The optimization schedule of the server-side hyper-prototypes should also be explained more clearly.
  - It should be clearer how many server-side update steps are performed per round and how those interact with the outer communication rounds.

- Scalability is not fully addressed.
  - The method maintains multiple hyper-prototypes per class and contrasts across classes.
  - It would be useful to discuss how well this scales to large numbers of classes and more realistic large-scale settings.

---

> ### Author Rebuttal · Authors · 2026-03-30
>
> Dear Reviewer fbGV, we appreciate your valuable time and constructive feedback, and we hope we addressed your concerns. Please let us know if you'd like any further information. Many thanks!
>
> **[W1,L6] The theoretical analysis seems closer to local objective**: We apologize for this confusion. Our analysis corresponds to the deviation bound and non-convex convergence rate of the objective $\mathcal{L}$ under the round-wise fixed hyper-prototypes. We would like to clarify that server-side optimization is not completely decoupled from our analysis (because the hyper-prototypes are used in $\mathcal{L}$); Sec. 3.3 analyzes the convergence of $\mathcal{L}$ conditioned on the server-optimized hyper-prototypes. We promise to revise the paper to state this scope explicitly.
>
> **[W2] The baseline comparison is not fully fair**: We agree the original protocols should be reported separately. The adapted variants, whose results are substantially better than those in the original protocols, are intended only for comparison under the same evaluation protocol. Besides Tab.A7, we now add original FedProto, FedTGP, and FedSA results on CIFAR10/CIFAR10-LT (10 clients, 10 local epochs), and will include them in Sec. 5.
> |Method|NID1$_{0.2}$|NID1$_{0.5}$|NID2|$\rho=10$|$\rho=50$|$\rho=100$|Comm.(MB)|
> |---|---|---|---|---|---|---|---|
> |FedProto|76.58|80.42|66.90|69.49|62.14|51.87|0.358×10=3.58|
> |FedTGP|79.18|82.70|70.53|72.55|65.36|56.25|0.358×10=3.58|
> |FedSA|78.72|83.11|71.75|73.02|65.73|57.60|0.361×10=3.61|
> |FedHPro|85.98|89.56|79.70|78.62|74.69|64.75|19.35×10=193.5|
>
> **[W3,L3,L4] The privacy discussion needs to be stronger**: This is an important point. FedHPro transmits class-wise average gradients for optimizing hyper-prototypes; it does not expose raw samples, and the server optimization is performed from statistics (with twice linear operations) without direct access to client data: such a design makes individual data reconstruction almost mathematically intractable [R1,R2]. We agree with the reviewer's concern under extreme label skew: we can respond to this by secure aggregation [R3]. For example, the gradients of clients $k_1, k_2$ as $\mathbf{g}\_{k_1}$ and $\mathbf{g}\_{k_2}$; client $k_1$ adds a random noise vector $\mathbf{I}$ to $\mathbf{g}\_{k_1}$ to get $(\mathbf{g}\_{k_1}+\mathbf{I})$, while client $k_2$ substitutes $\mathbf{I}$ from $\mathbf{g}\_{k_2}$ to get $(\mathbf{g}\_{k_2}-\mathbf{I})$. The two modified gradients are then sent to the server to perform Eq.(6) in the paper, i.e., $(\mathbf{g}\_{k_1}+\mathbf{I}+\mathbf{g}\_{k_2}-\mathbf{I})/2=(\mathbf{g}\_{k_1}+\mathbf{g}\_{k_2})/2$. We will revise our paper to include these discussions.
>
> [R1] Du, Jiacheng, et al. Sok: On gradient leakage in federated learning. USENIX Security 2025
>
> [R2] Pan, Xudong, et al. Exploring the security boundary of data reconstruction via neuron exclusivity analysis. USENIX Security 2022
>
> [R3] Segal, Aaron, et al. Practical secure aggregation for privacy-preserving machine learning. CCS 2017
>
> **[W4] Some implementation details should be improved**: We will move the method section earlier and emphasize the pseudocode. For server-side optimization, we initialize $\mathcal{S}_M$ as random noise and set the number of them per class at $|\mathcal{I}|$, and we use SGD with a learning rate 0.1 to optimize $\mathcal{S}_M$ over $M$ iterations per global round. We will clarify this near Eq.(9) and note the recommended values in Sec. 5.3 (e.g., $M=30$).
>
> **[W5,L5] Scalability is not fully addressed**: FedHPro uses $|\mathcal{I}|=5$ by default; with fixed $|\mathcal{I}|$, complexity is approximately linear in the number of classes $\mathbb{C}$. Empirically, FedHPro remains strong on CIFAR100 ($\mathbb{C}=100$), TinyImageNet ($\mathbb{C}=200$), long-tailed variants, and varying client numbers $K$ (Tab.A5). We also add experiments on SUN397 [R4] ($\mathbb{C}=397$, $K=100$, 20% active clients), supporting applicability to larger-scale settings.
> |Settings|FedAvg|FedRCL|FedSA|FedSA$\dagger$|FedHPro|
> |---|---|---|---|---|---|
> |NID1$_{0.2}$|68.92|70.23|61.58|70.86|73.41|
> |NID1$_{0.5}$|70.61|73.26|65.30|72.18|75.22|
>
> [R4] Xiao, Jianxiong, et al. Sun database: Large-scale scene recognition from abbey to zoo. CVPR 2010
>
> **[L1] The restricted nature of the model-heterogeneity setting**: The model-heterogeneity setting is a restricted protocol: clients use different feature extractors, with only the classifier globally aggregated; each client keeps its own feature extractor locally. This is because FedHPro relies on the global classifier to obtain gradients of hyper-prototypes via a virtual loss in Eq.(7).
>
> **[L2] The dependence on classifier and matched feature dimension**: FedHPro requires cross-client feature comparability in the same latent space; otherwise, the similarity in HPCL (Eq.(11)) and the alignment in HPAL (Eq.(13)) are not well-defined. So the clients (if they use different feature extractors) need to make the output dimension consistent.

---

> > ### Author Rebuttal · Reviewer_fbGV · 2026-04-02
> >
> > Thank you for the detailed rebuttal. It addresses several of my main concerns, especially by clarifying the privacy implications of the shared gradients and by improving the fairness of the baseline comparison through additional original-protocol results and stronger experimental evidence. I still think some theoretical aspects could be clarified further, particularly the scope of the convergence analysis, but overall the rebuttal strengthens the paper enough that I am now comfortable supporting acceptance. Accordingly, I am increasing my score.

---

> > > ### Author Response · Authors · 2026-04-03
> > >
> > > Dear Reviewer fbGV, thank you for your thoughtful response.
> > >
> > > We are genuinely grateful for your constructive feedback. We are immensely appreciative of the discussions on the privacy of the shared gradients and the theoretical analysis, as they clearly help us improve our paper and strengthen future extensions.
> > >
> > > Best,
> > >
> > > the Authors

---

### Decision · Program_Chairs · 2026-04-30

**Decision:**

Accept (regular)

**Comment:**

This paper proposes a prototype-based federated learning method that replaces a single averaged global prototype per class with a set of learnable class-wise hyper-prototypes optimized on the server via gradient matching. The problem of semantic drift of class prototypes under heterogeneous client distributions is important in FL. The core idea is interesting and makes sense.